# Burst mitofusin activation reverses neuromuscular dysfunction in murine CMT2A

Antonietta Franco[1†], Xiawei Dang[1,2†], Emily K Walton[1], Joshua N Ho[3,4], Barbara Zablocka[5], Cindy Ly[6], Timothy M Miller[6], Robert H Baloh[7], Michael E Shy[8], Andrew S Yoo[3,4], Gerald W Dorn II[1]*

[1]Department of Internal Medicine, Pharmacogenomics, Washington University School of Medicine, St Louis, United States; [2]Department of Cardiology, The First Affiliated Hospital of Xi'an Jiao Tong University, Xi'an, China; [3]Department of Developmental Biology, Washington University School of Medicine, St Louis, United States; [4]Center for Regenerative Medicine, Washington University School of Medicine, St. Louis, United States; [5]Mossakowski Medical Research Centre, Polish Academy of Sciences, Warsaw, Poland; [6]Department of Neurology, Washington University School of Medicine, St Louis, United States; [7]Department of Neurology, Cedars-Sinai Medical Center, Los Angeles, United States; [8]Department of Neurology, Carver College of Medicine, University of Iowa, Iowa City, United States

**Abstract** Charcot–Marie-Tooth disease type 2A (CMT2A) is an untreatable childhood peripheral neuropathy caused by mutations of the mitochondrial fusion protein, mitofusin (MFN) 2. Here, pharmacological activation of endogenous normal mitofusins overcame dominant inhibitory effects of CMT2A mutants in reprogrammed human patient motor neurons, reversing hallmark mitochondrial stasis and fragmentation independent of causal *MFN2* mutation. In mice expressing human *MFN2* T105M, intermittent mitofusin activation with a small molecule, MiM111, normalized CMT2A neuromuscular dysfunction, reversed pre-treatment axon and skeletal myocyte atrophy, and enhanced axon regrowth by increasing mitochondrial transport within peripheral axons and promoting in vivo mitochondrial localization to neuromuscular junctional synapses. MiM111-treated *MFN2* T105M mouse neurons exhibited accelerated primary outgrowth and greater post-axotomy regrowth, linked to enhanced mitochondrial motility. MiM111 is the first pre-clinical candidate for CMT2A.

*For correspondence:
gdorn@wustl.edu

†These authors contributed equally to this work

## Introduction

Charcot–Marie-Tooth disease (CMT) describes a family of genetically diverse and clinically heterogeneous peripheral neuropathies (*Fridman et al., 2015*). Type 2A CMT (CMT2A) is caused by mutations of the mitochondrial fusion protein, mitofusin 2 (*MFN2*) (*Züchner et al., 2004*), and is distinguished from other CMT subtypes by onset of neuromuscular signs in early childhood and progressive loss of neuromuscular coordination and strength in arms throughout the first two decades of life, thought to be the consequence of dying-back of long peripheral nerves (*Fridman et al., 2015*; *Feely et al., 2011*; *Bombelli et al., 2014*; *Yaron and Schuldiner, 2016*; *Berciano et al., 2017*). Because there are currently no disease-modifying treatments, CMT2A is managed with braces, wheelchairs, and social support.

Over 100 different dominant missense *MFN2* mutations are implicated in CMT2A (*Bereşewicz et al., 2018*). MFN2 and related MFN1 are nuclear-encoded dynamin-family GTPases

**eLife digest** Charcot-Marie-Tooth disease type 2A is a rare genetic childhood disease where dying back of nerve cells leads to muscle loss in the arms and legs, causing permanent disability. There is no known treatment.

In this form of CMT, mutations in a protein called mitofusin 2 damage structures inside cells known as mitochondria. Mitochondria generate most of the chemical energy to power a cell, but when mitofusin 2 is mutated, the mitochondria are less healthy and are unable to move within the cell, depriving the cells of energy. This particularly causes problems in the long nerve cells that stretch from the spinal cord to the arm and leg muscles.

Now, Franco, Dang et al. wanted to see whether re-activating mitofusin 2 could correct the damage to the mitochondria and restore the nerve connections to the muscles. The researchers tested a new class of drug called a mitofusin activator on nerve cells grown in the laboratory after being taken from people suffering from CMT2A, and also from a mouse model of the disease. Mitofusin activators improved the structure, fitness and movement of mitochondria in both human and mice nerve cells. Franco, Dang et al. then tested the drug in the mice with a CMT2A mutation and found that it could also stimulate nerves to regrow and so reverse muscle loss and weakness.

This is the first time scientists have succeeded to reverse the effects of CMT2A in nerve cells of mice and humans. However, these drugs will still need to go through extensive testing in clinical trials before being made widely available to patients. If approved, mitofusin activators may also be beneficial for patients suffering from other genetic conditions that damage mitochondria.

located at the mitochondrial outer membrane-cytosol interface where they promote mitochondrial fusion essential to mitochondrial respiratory function and repair (*Chan, 2012*). Dominant inhibition by MFN2 mutants of mitochondrial fusion (*Chen and Chan, 2006*; *Pareyson et al., 2015*), mitophagy (*Rizzo et al., 2016*; *Filadi et al., 2018*), and/or neuronal mitochondrial transport (*Pareyson et al., 2015*; *Baloh et al., 2007*; *Crunkhorn, 2018*) are proposed to evoke neuronal degeneration in CMT2A.

Because CMT2A is an autosomal dominant genetic condition, gene editing could potentially correct causal *MFN2* mutant alleles, but the large number of different causal CMT2A *MFN2* mutations complicates an editing approach. Alternately, forced expression of normal mitofusins could oppose mutant *MFN2* dysfunction, as demonstrated in a recent study in transgenic mice (*Zhou et al., 2019*). However, MFN gene therapy would be difficult to discontinue or reverse if postulated adverse effects of MFN overactivity are encountered (*El Fissi et al., 2018*). Here, we describe a therapeutic approach to CMT2A that is agnostic to *MFN2* genotype and does not require genetic manipulation: intermittent or 'burst' activation of endogenous normal mitofusins. Pharmacological mitofusin activators improved mitochondrial morphology, fitness, and motility in human and mouse CMT2A neurons in vitro. Daily administration of a short acting mitofusin activator to mice with late stage CMT2A reversed neuromuscular dysfunction. Mechanistically, neuronal repair and regeneration were linked to enhanced mitochondrial transport to, and mitochondrial occupation within, axonal termini. Reversal of pre-existing CMT2A neuromuscular degeneration in vivo has not previously been achieved by any means, and provides a powerful rationale for advancing mitofusin activators to first in human trials.

## Results

### Genetically diverse CMT2A patient neurons exhibit similar mitochondrial phenotypes

One of the central features of CMT2A is the large number of different *MFN2* mutations that provoke the syndrome. Common MFN2 GTPase and coiled-coiled domain mutations induce more severe and earlier onset disease, whereas rare carboxy terminal domain mutations confer later onset and milder disease (*Feely et al., 2011*; *Stuppia et al., 2015*). We compared mitochondrial phenotypes in cells from four CMT2A patients, two having *MFN2* mutations within the canonical dynamin/Fzo-like GTPase domain (*MFN2* T105M in the G1 motif and *MFN2* R274W between the G4 and G5 motifs),

and two with mutations in the *MFN2* coiled-coiled helix bundle core (*MFN2* H361Y and R364W). (*Figure 1—figure supplement 1*). Donor patient characteristics are in *Table 1*.

To avoid loss of some CMT2A-associated mitochondrial phenotypes in iPSC-derived neurons (*Rizzo et al., 2016*; *Saporta et al., 2015*), we directly reprogrammed CMT2A patient fibroblasts into motor neurons via microRNA-mediated neuronal conversion (*Figure 1b*; *Abernathy et al., 2017*). Reprogramming efficiency was similar between CMT2A and control patient fibroblasts:>90% neurons (measured as β-III tubulin staining), and >85% motor neurons (measured as β-III tubulin, HB9/MNX1 co-staining) (*Figure 1—figure supplement 2*). Compared to neurons reprogrammed from individuals with no evident disease at the time of sampling and who had none of the tested MFN2 mutations by Sanger sequencing ('normal'), all four CMT2A motor neuron lines exhibited fragmented mitochondria (decreased mitochondrial aspect ratio; length/width) that is a consequence of impaired fusion in this context *Franco et al., 2016*; accompanying mitochondrial depolarization reflected characteristic functional impairment (*Figure 1c*; *Crowley et al., 2016*). Moreover, all four CMT2A motor neuron lines exhibited abnormal mitochondrial transport through axons, with diminished proportion and velocity of motile mitochondria (*Figure 1c*). Mitochondrial fragmentation, respiratory dysfunction, and dysmotility observed in reprogrammed neurons are prototypical features of CMT2A (*Baloh et al., 2007*; *Zhou et al., 2019*; *Verhoeven et al., 2006*; *Rocha et al., 2018*).

Dominant inhibition of normal *MFN1* and *MFN2* by CMT2A *MFN2* mutants produces an imbalance between mitochondrial fission and fusion that underlies mitochondrial pathology in CMT2A (*Zhou et al., 2019*). This dynamic imbalance can be reversed in transfected mouse cells and in vivo mouse models by forced overexpression of normal MFN1 or MFN2 (*Zhou et al., 2019*; *Detmer and Chan, 2007*). We posited that pharmacological activation of normal endogenous human *MFN1* and *MFN2* would also reverse mitochondrial abnormalities in CMT2A patient motor neurons. Chimera C is one of a new class of direct mitofusin activators that promotes conformational activation of MFN1 and MFN2, thereby stimulating endogenous mitofusins to improve mitochondrial dysmorphology and dysfunction (*Rocha et al., 2018*; *Dang et al., 2020*). Chimera C (100 nM, 48 hr) enhanced mitochondrial fusion (i.e. it increased aspect ratio) and improved respiratory function (i.e. it reversed mitochondrial depolarization) in cells lacking either MFN1 or MFN2, but had no effects in cells lacking both mitofusin targets (*Figure 1—figure supplement 3*). Chimera C (100 nM, 48 hr) also improved mitochondrial aspect ratio, depolarization, and motility in all four CMT2A patient motor neuron lines (*Figure 1c*).

## Neuron-specific expression of MFN2 T105M in mice recapitulates key features of human CMT2A

Children with CMT2A are typically healthy during early years, but develop signs of neuromuscular dysfunction during the mid first decade of life. Neurogenic distal limb muscular atrophy is progressive until the end of the second decade, at which time the disease stabilizes; longevity is normal, but

**Table 1.** Characteristics and sources of human primary fibroblasts used for motor neuron reprogramming studies.

| Diseases | Mutation | Age | Sex | Passage# | Source | Fibroblast ID |
|---|---|---|---|---|---|---|
| CMT2A | *MFN2* Thr105Met | 41 | F | P4-P10 | Dr. Robert H. Baloh | - |
| CMT2A | *MFN2* Arg274Trp | 23 | M | P4-P10 | Dr. Barbara Zablocka | - |
| CMT2A | *MFN2* His361Tyr | 41 | M | P4-P10 | Dr. Robert H. Baloh | - |
| CMT2A | *MFN2* His364Trp | 28 | F | P6-P10 | Dr. Michael E. Shy | - |
| CMT1A | *PMP22* DUP | 28 | F | P4-P10 | Coriell Institute | GM05167 |
| CTRL 1 | - | 68 | F | P3-P7 | NINDS | ND34769 |
| CTRL 2 | - | 71 | F | P3-P7 | NINDS | ND36320 |
| CTRL 3 | - | 55 | F | P3-P7 | NINDS | ND29510 |
| CTRL 4 | - | 66 | M | P8-P10 | NINDS | ND29178 |
| CTRL 5 | - | 72 | M | P3-P7 | NINDS | ND34770 |
| CTRL 6 | - | 55 | M | P4-P10 | NINDS | ND38530 |

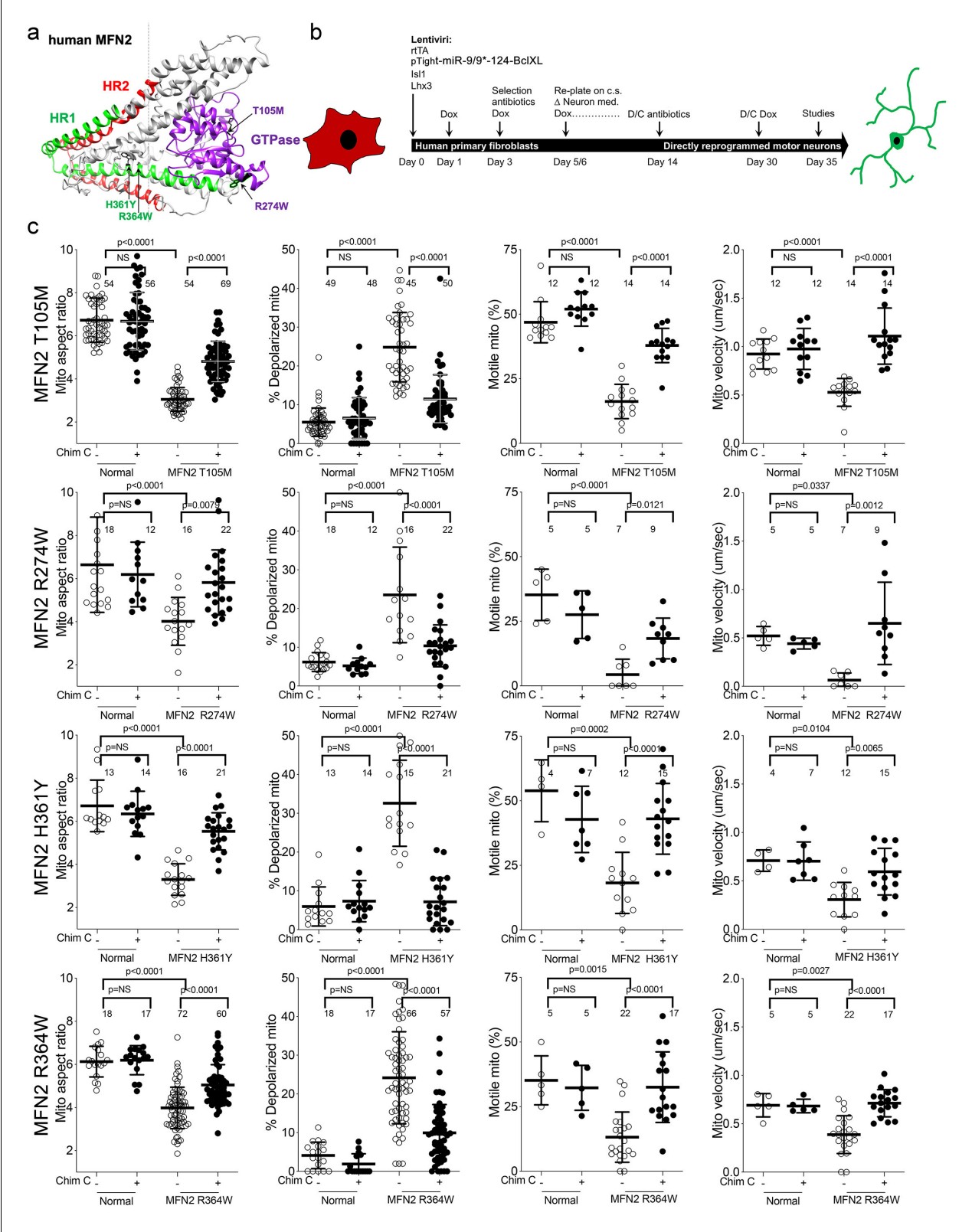

**Figure 1.** Mitochondrial abnormalities in reprogrammed CMT2A patient motor neurons and their improvement after mitofusin activation. (**a**) Model structure of *MFN2* showing location of CMT2A patient mutations. (**b**) Schematic depiction of fibroblast reprogramming procedure to produce motor neurons. (**c**) Mitochondrial testing in reprogrammed motor neurons from four CMT2A patients with different MFN2 mutations and representative of

*Figure 1 continued on next page*

*Figure 1 continued*

three normal control subjects. Open circles are baseline; closed circles are 48 hr after addition of mitofusin activator Chimera C (100 nM). Each circle is one neuron from two or three independent reprogrammings. P values from ANOVA.

The online version of this article includes the following figure supplement(s) for figure 1:

**Figure supplement 1.** Genotyping of CMT2A patient cells.
**Figure supplement 2.** Direct reprogramming of human skin fibroblasts to neurons.
**Figure supplement 3.** Chemical characteristics and functional profiling of mitofusin activators used in this study.

disability is permanent (*Fridman et al., 2015*; *Feely et al., 2011*). No mouse models of CMT2A recapitulate all of these key clinical features in the absence of confounding developmental phenotypes (*Zhou et al., 2019*; *Detmer et al., 2008*; *Cartoni et al., 2010*; *Bannerman et al., 2016*; *Dorn, 2020*). Therefore, a prerequisite for proof-of-concept testing of mitofusin activation in vivo was to generate a mouse CMT2A model having greater similarity to the human condition.

By combining Rosa26 <fs-^MFN2(T105M)^> (*Bannerman et al., 2016*) and *Mnx1-Cre* (HB9) (*Yang et al., 2001*) alleles (*Figure 2a*) we drove human *MFN2* T105M expression in mouse neurons (*Figure 2b*; CMT2A mouse). Neuromuscular functional integrity over time was assessed as the duration mice could walk on an elevated accelerating rotating cylinder without falling off (RotaRod latency). RotaRod latency of CMT2A mice was normal at 10 weeks of age, progressively declined thereafter, and stabilized beyond 30 weeks (*Figure 2c*). As in clinical CMT2A, axonal mitochondria of MFN2 T105M mice were fragmented with disorganized cristae (*Sole et al., 2009*; *Figure 2d*).

Neuroelectrophysiological testing of CMT2A patients characteristically reveals reduced compound muscle action potentials (CMAP) with normal nerve conduction velocities (*Berciano et al., 2017*; *Harding and Thomas, 1980*). Recapitulating this clinical finding, sciatic nerve-tibialis muscle CMAP amplitudes of 50-week-old MFN2 T105M mice were diminished with no change in signal latency, which reflects conduction velocity (*Figure 2e*). Tibialis myofiber atrophy and loss of large axons without demyelination in the MFN2 T105M mouse (*vide infra*) also mimicked clinical CMT2A (*Verhoeven et al., 2006*; *Muglia et al., 2001*; *Neves and Kok, 2011*).

To further evaluate the relevance of the *MFN2* T105M mouse to human CMT2A, dorsal root ganglion (DRG) sensory neurons were isolated and placed in culture, the MFN2 T105M transgene induced with Adeno-Cre, and neurons assayed for the mitochondrial pathologies delineated in reprogrammed CMT2A patient motor neurons (*vide supra*). CMT2A-associated abnormalities in axon mitochondrial aspect ratio and transport (*Figure 2f*) and polarization status (*Figure 2—figure supplement 1*) were each mimicked in mouse CMT2A DRGs. As in reprogrammed human CMT2A motor neurons, mitofusin activation improved these abnormalities (*Figure 2f* and *Figure 2—figure supplement 1*, compare to *Figure 1c*).

## Burst mitofusin activation reverses neuromuscular dysfunction in CMT2A mice

Collectively, the above results show that activating mitofusins can improve multiple mitochondrial abnormalities manifested by cultured human and mouse CMT2A neurons. To determine whether benefits of mitofusin activation in cultured neurons would translate to therapeutic effects on neuromuscular dysfunction in CMT2A we contemplated an in vivo trial in our CMT2A mouse. However, Chimera C is rapidly degraded by the liver and undergoes first-pass metabolism, making it impractical for in vivo studies (*Dang et al., 2020*). We therefore evaluated in vivo efficacy of mitofusin activation in CMT2A using MiM111, a structurally distinct compound having a mitofusin activation profile similar to Chimera C (*Figure 1—figure supplement 3*), but which is metabolically stable with good nervous system bioavailability (*Dang et al., 2020*). We hypothesized that intermittent or 'burst' mitofusin activation (a dosing schedule that reversed mitochondrial dysfunction for <12 hr each day) (*Figure 3—figure supplement 1*) would confer therapeutic benefits by cyclically enhancing mitochondrial fitness and transport, while minimizing the possibility of mitofusin toxicity that might occur with constant mitofusin activation (*El Fissi et al., 2018*; *Meyer et al., 2017*).

Based on a minimum effective MiM111 plasma concentration of 30 ng/ml (*Dang et al., 2020*) and a plasma $t_{1/2}$ of 2.3 hr with $C_{max}$ of 24,000 ng/ml after intramuscular administration of 30 mg/kg (*Figure 3—figure supplement 1*), we estimated that daily IM dosing would reverse mitochondrial

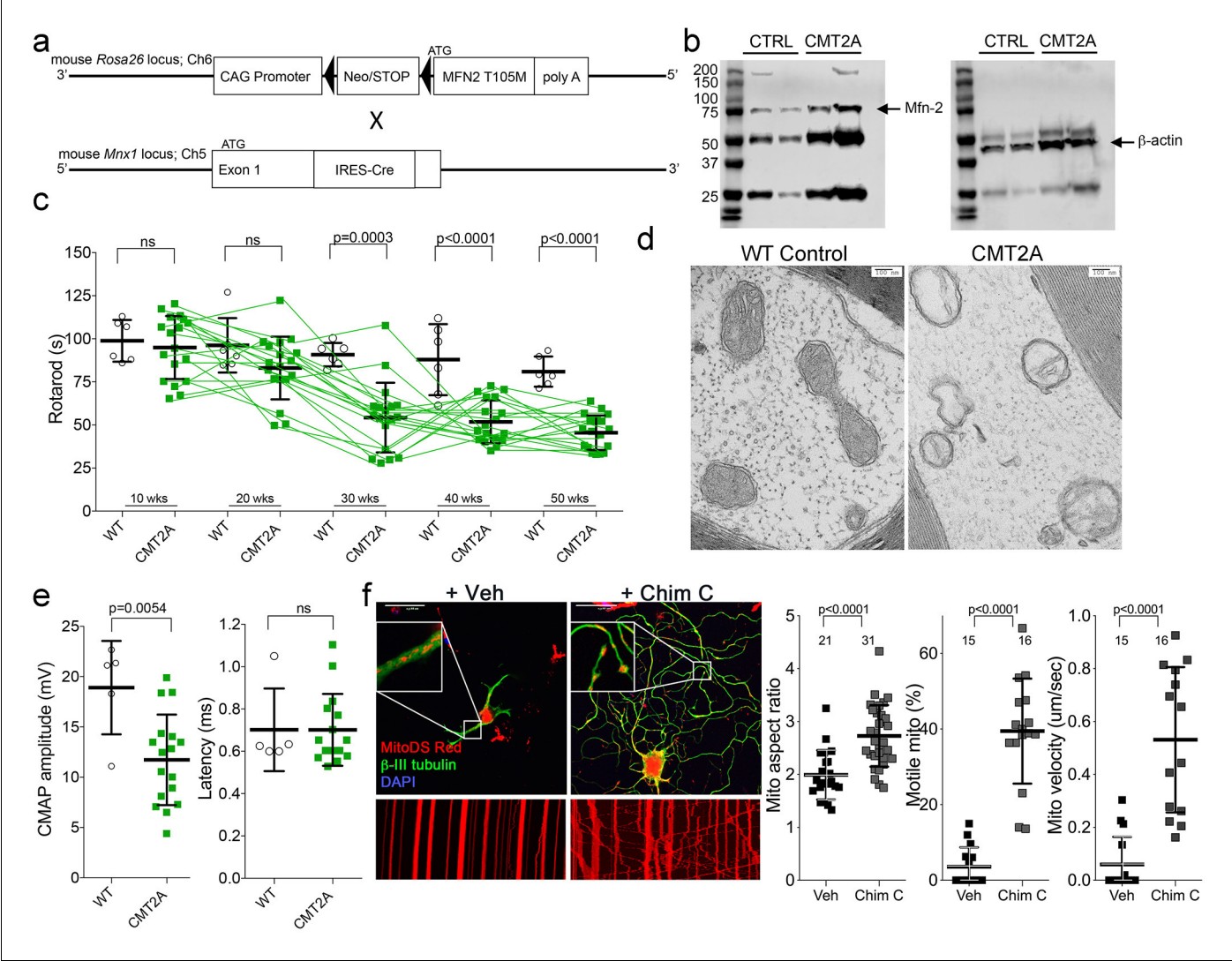

**Figure 2.** Characteristics of a neuron-specific MFN2 T105M mouse model of CMT2A. (**a**) Schematic depiction Mfn2 <sup><fs-T105M></sup> expression strategy. (**b**) Immunoblot analysis of MFN2 expression in mouse sciatic nerves. (**c**) Serial RotaRod latency studies; CMT2A is green squares (n = 16), wild-type (WT) control is open circles (n = 6). (**d**) Electron micrographs of axonal mitochondrial from sciatic nerves (50 weeks). (**e**) Comparative neuro-electrophysiology study results of 50-week-old mice in panel **c**. (**f**) Response of CMT2A dorsal root ganglion neurons to mitofusin activation with Chimera C (100 nM, 48 hr). Top images are confocal micrographs of DRGs stained for mitochondria (red) and axons (green). Insets are higher power magnification to see mitochondrial morphology. Bottom images are kymographs showing mitochondrial (red) motility. Vertical columns are stationary mitochondria; lines transiting left to right or right to left are moving. P values are from t-test from 3 or four independent experiments.

The online version of this article includes the following figure supplement(s) for figure 2:

**Figure supplement 1.** Flow cytometric profiling of mitochondrial polarization status in mouse dorsal root ganglion (DRG) neurons.

dysmotility in CMT2A mice for ~12 hr out of every 24 hr. Indeed, *Figure 3—figure supplement 1* show that mitochondrial motility in sciatic nerve axons of MFN2 T105M mice was normalized 4 hr after a single intramuscular injection of MiM111 (30 mg/kg), declined by approximately half after 12 hr, and returned to CMT2A baseline after 24 hr.

If CMT2A neuron die-back is reversible then burst mitofusin activation should improve neuromuscular degeneration in MFN2 T105M mice who had progressed to the severe and stable CMT2A phenotype. To test this notion, 50-week-old MFN2 T105M mice and littermate controls were randomized to receive daily MiM111 or its vehicle. Researchers blind to genotype and treatment group performed Rotarod and neuro-electrophysiological testing after 4 and 8 weeks (*Figure 3b*).

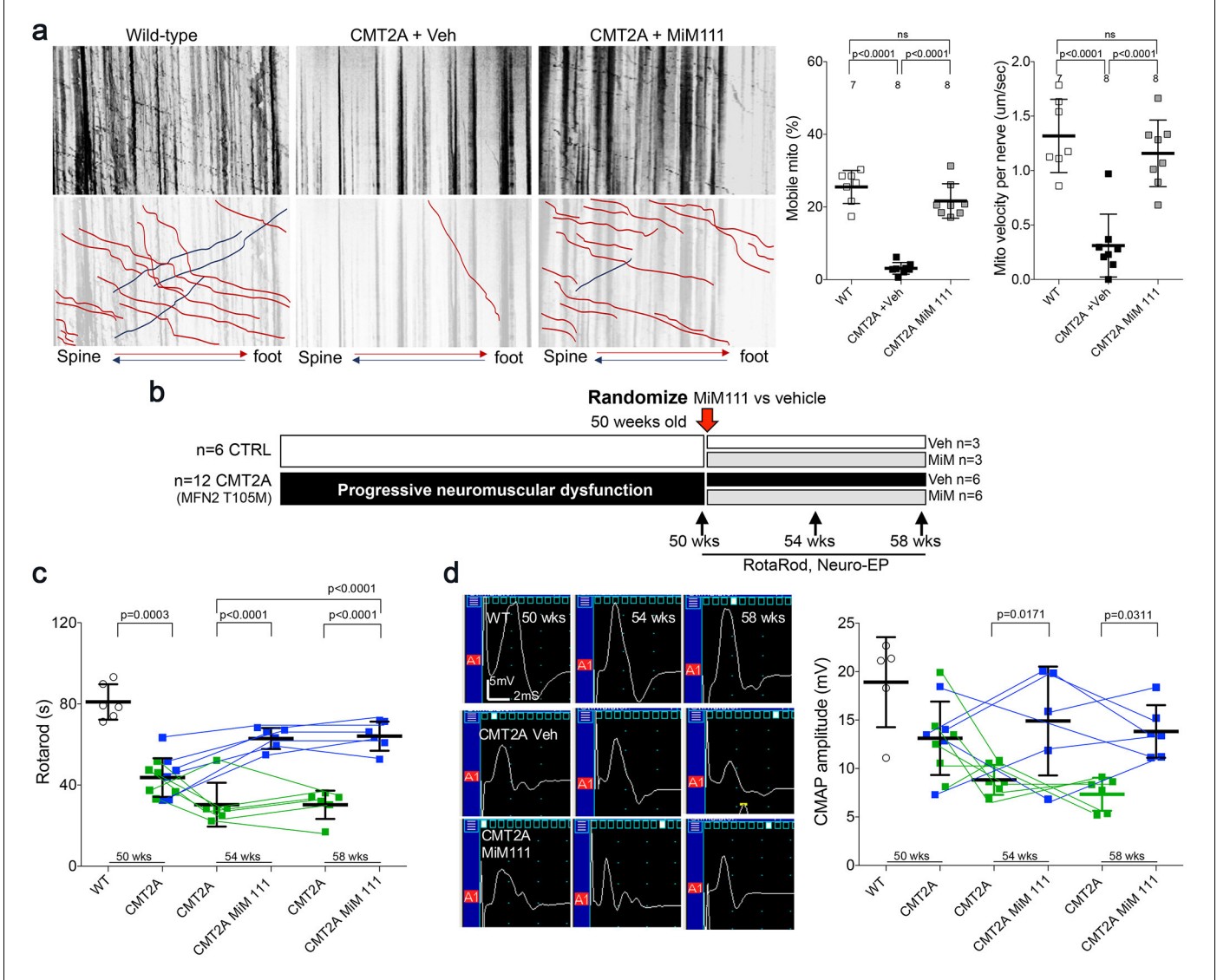

**Figure 3.** Mitofusin activation reverses neuromuscular dysfunction in MFN2 T105M mice. (**a**) Ex vivo mitochondrial motility in CMT2A mouse sciatic nerve axons 4 hr after intramuscular administration of mitofusin activator MiM111 or vehicle. Top panel is kymographs. Bottom panel emphasizes motile mitochondria with red and blue lines transiting antegrade or retrograde, respectively. (Note, mitochondrial transport in ex vivo sciatic nerves favors the antegrade [spine to foot] direction because mitochondria are recruited to the site of nerve injury at the distal amputation site [*Zhou et al., 2016*]). (**b**) Experimental design to evaluate efficacy of MiM111 in late murine CMT2A. (**c**) RotaRod latency in vehicle- (green) and MiM111-treated (blue) MFN2 T105M mice. (**d**) Neuroelectrophysiology studies: (left) representative CMAP tracings; (right) quantitative data. Each symbol in c and d is one mouse. P values from ANOVA. WT control values are open circles in panels c and d; complete WT control data are in *Figure 3—figure supplement 2*. The online version of this article includes the following figure supplement(s) for figure 3:

**Figure supplement 1.** In vivo pharmacokinetics and target engagement of MiM111 administered intramuscularly.

**Figure supplement 2.** Effects of MiM111 on neuromuscular function in control mice.

The characteristic decreases in RotaRod latency and CMAP amplitude in MFN2 T105M mice (see *Figure 2c and e*) were reversed 4 weeks after MiM111 treatment and remained near normal after 8 weeks (*Figure 3c and d*); MiM111 had no effect on control mice (*Figure 3—figure supplement 2*).

Compared to sciatic nerves of vehicle-treated MFN2 T105M CMT2A mice, MiM111 treatment reduced axon damage (*Figure 4a*), increased axon diameter (*Figure 4b*), and increased staining for superior cervical ganglion 10 (SCG10; a marker of neuron regeneration) (*Shin et al., 2014*;

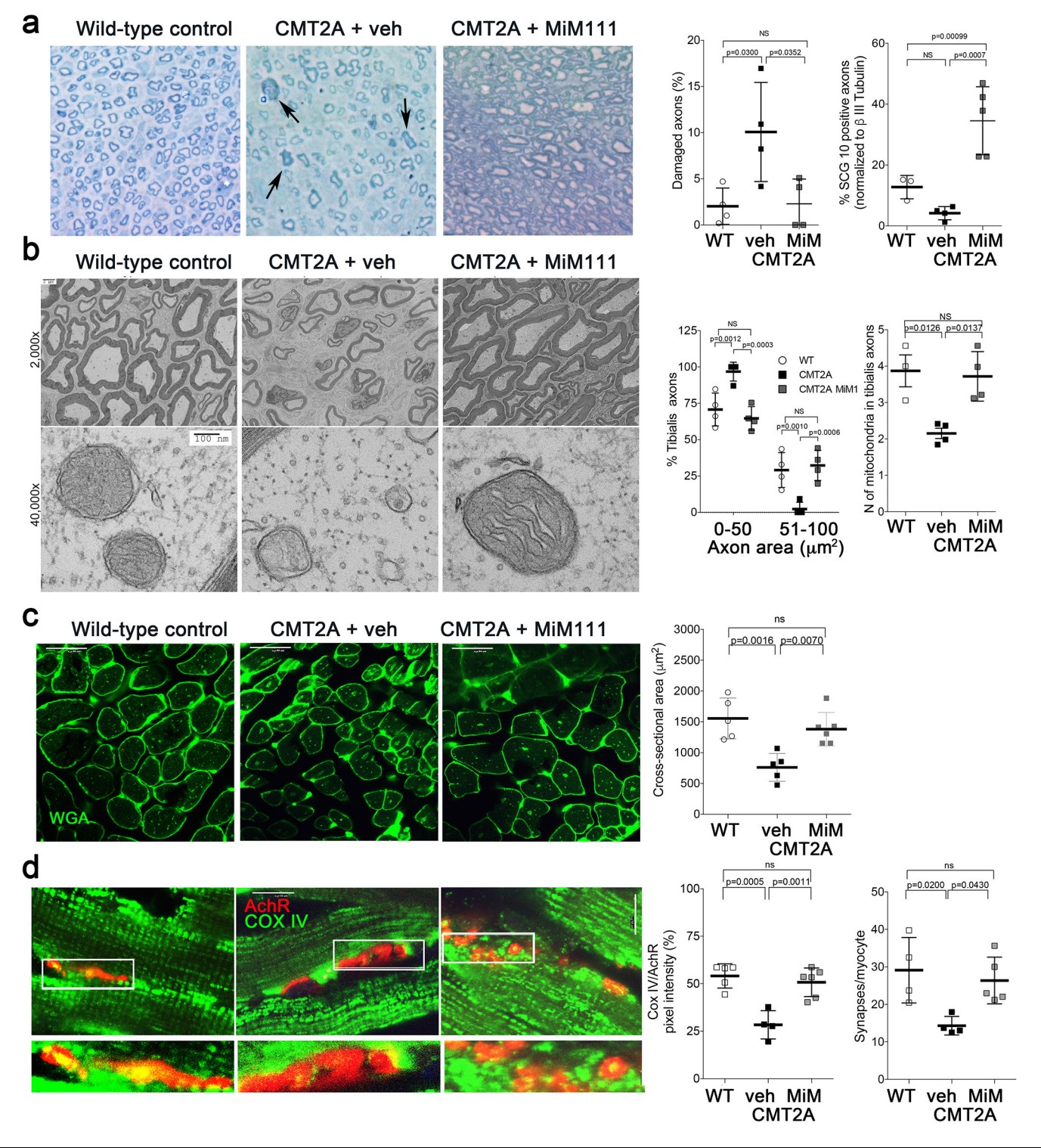

**Figure 4.** Mitofusin activation reverses histopathological findings in MFN2 T105M mice. (**a**) Toluidine blue stained sections of mouse mid tibial nerves. Arrows show blue-stained damaged axons in CMT2A mice. Quantitative group data for damaged axons and SCG10-regenerating axons (see *Figure 4—figure supplement 1*) are on the right. (**b**) Electron micrographs of mid-tibial nerve axons from CMT2A mouse study groups after 8 weeks of therapy. Note heterogeneity in axon size (top images; left graph) and mitochondrial abnormalities (bottom images, right graph). (**c**) Wheat germ agglutinin (WGA) labeled sections of tibialis anterior muscle and quantitative myocyte cross sectional area. (**d**) Confocal micrographs of neuromuscular junctions to show mitochondrial occupancy yellow organelles within red synapses (also see *Figure 4—figure supplement 1*). Each symbol represents results from one mouse. Data are means ± SD; p values are 1- or 2-way ANOVA.

*Figure 4 continued on next page*

*Figure 4 continued*

The online version of this article includes the following figure supplement(s) for figure 4:

**Figure supplement 1.** Effects of MiM111 on neuromuscular integrity in CMT2A MFN2 T105M mice.

*Figure 4—figure supplement 1*). These findings suggest that mitofusin activation reversed CMT2A-induced neuronal degeneration.

Skeletal myocytes of CMT2A mouse tibialis muscle innervated by the sciatic nerve were abnormally small (*Figure 4c*), reflecting neurogenic muscle atrophy (because the MFN2 T105M transgene is directed by neuron-specific HB9-Cre). In agreement with muscle atrophy being a secondary effect, skeletal myocyte mitochondria of CMT2A mice appeared structurally normal (*Figure 4—figure supplement 1*) compare to sciatic nerve axon mitochondria in *Figures 2d* and *4b*. Therefore, normalization of tibialis myocyte diameter after mitofusin activation (*Figure 4c*) indicates restoration of neuromuscular integrity.

Collectively, the above findings provide indirect support for the idea that CMT2A mice suffer from distal neuron dieback that can be reversed by activating mitofusins. Reasoning that decreased neuromuscular junction density in CMT2A mice would constitute direct evidence for dieback, we quantified neuromuscular junctional synapses containing receptors for the neurotransmitter acetylcholine (AchR) in tibialis muscles of CMT2A mice. Compared to normal mice, CMT2A mice had ~50% fewer synaptic junctions/myocyte, which was reversed after mitofusin activator treatment (*Figure 4—figure supplement 1*). Strikingly, mitochondrial occupancy of vehicle-treated CMT2A neuromuscular synaptic junctions was also reduced by ~half compared to normal mice, and was normalized by MiM111 treatment (*Figure 4d*). Because mitochondrial transport can play a central role in neuron repair and regeneration (*Sheng, 2017*), the observation that MiM111 promoted mitochondrial localization within terminal neuromuscular synaptic junctions provided a plausible mechanistic link between mitofusin activation, mitochondrial motility, neuronal regrowth, and reversal of neuromuscular dysfunction in this preclinical CMT2A model.

## Enhanced mitochondrial function in mitofusin-activated CMT2A DRGs leads to accelerated axon growth

Reversal of CMT2A-induced distal neuron die back implies neuronal regrowth. Indeed, sensory DRG neurons isolated from CMT2A mice and cultured in the presence of MiM111 (100 nM, 48 hr) exhibited not only enhanced mitochondrial fusion (increased aspect ratio) and transport (greater mitochondrial motility and velocity), but axon outgrowth (length and branching) (*Figure 5*). Similar effects were seen in CMT2A DRGs treated with Chimera C (100 nM, 48 hr) (*Figure 5—figure supplement 1*). Both MiM111 and Chimera C provoked mitochondrial redistribution to axonal termini of cultured CMT2A DRGs (*Figure 5—figure supplement 1*) recapitulating mitochondrial occupation of neuromuscular synapses after MiM111 treatment of CMT2A mice in vivo (see *Figure 4d*).

Comparing the mitochondrial motility, aspect ratio, and neuron growth responses at different times after mitofusin activation revealed significantly increased mitochondrial trafficking within 2 hr, whereas enhanced axon outgrowth was significant after 24 and 28 hr, and mitochondrial aspect ratio (i.e. fusion) was significant only after 48 hr (*Figure 5—figure supplement 2*). Given the established role for mitochondrial transport in neuronal repair (*Sheng, 2017*), this temporal sequence lends credence to the idea that accelerated neuron growth is a consequence of enhanced mitochondrial function and redistribution.

## Mitofusin activation accelerates in vitro CMT2A axon regeneration after axotomy

DRG outgrowth measures in vitro regrowth of neuronal extensions that are amputated during the cell isolation trituration procedure. We considered that a more appropriate model of regrowth after dieback in CMT2A would test intact neurons lacking only the distal axons. Because CMT2A mouse neurons grow poorly in tissue culture in the absence of mitofusin activators (*vide supra*), this was not feasible using DRGs. Therefore, we seeded cortical neurons collected from MFN2 T105M allele mice in chambers separated from empty chambers by linear microchannels. In the absence of Cre-recombinase these 'normal' neurons grew axons through the microchannels that branched into the empty

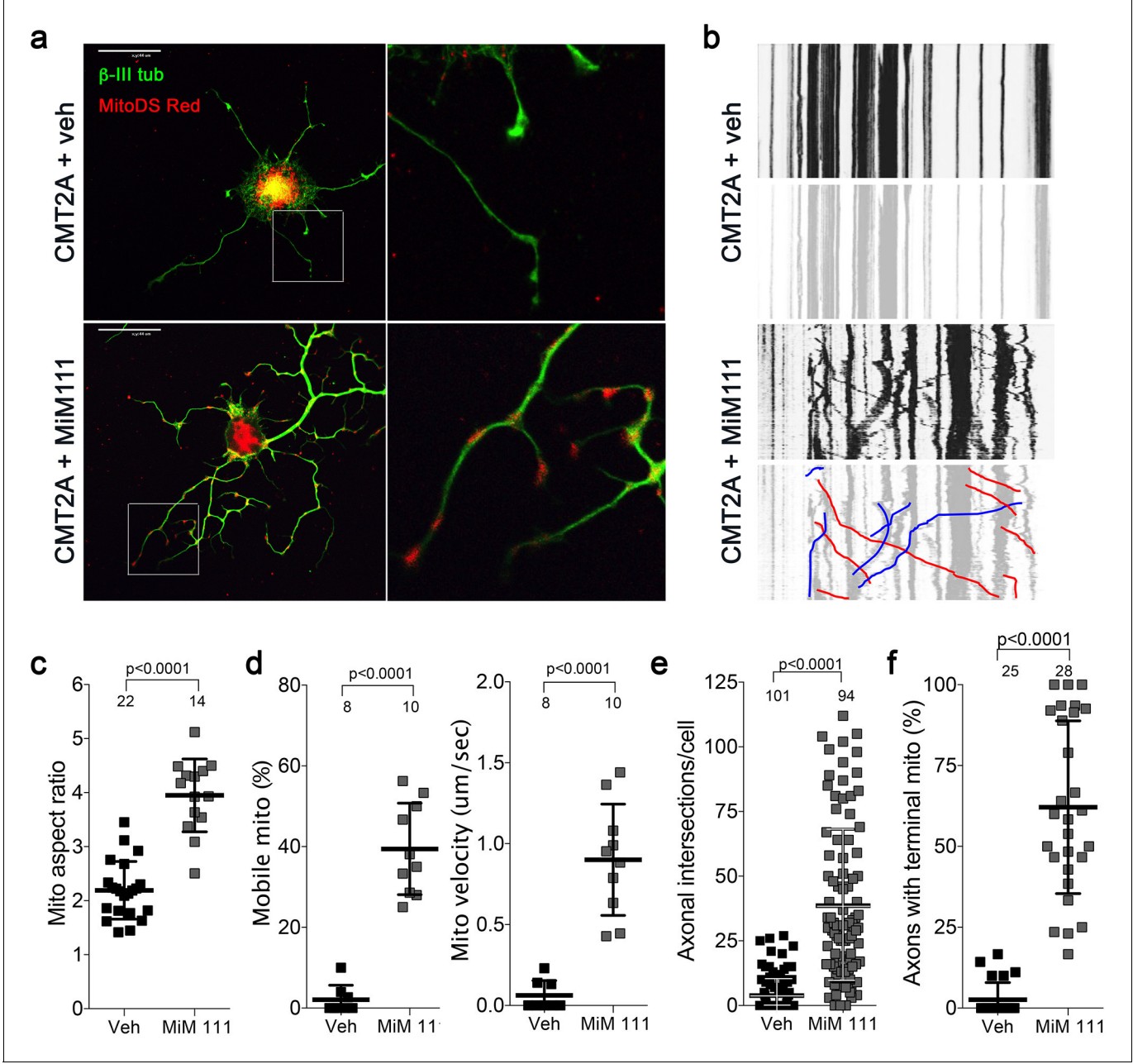

**Figure 5.** Mitofusin activation reverses mitochondrial pathology and stimulates growth of CMT2A dorsal root ganglion neurons in vitro. (a) Confocal micrographs of CMT2A mouse DRGs cultured for 48 hr with MiM111 or its vehicle. Note greater neuronal process length and branching in MiM111-treated neuron. Exploded insets (right) show neuronal process termini. Mitochondria express mitoDS Red; neuronal processes stained for β-III tubulin are green. (b) Kymographs of mitochondrial motility in neuronal processes of live DRGs from studies shown in (a). Top panel is raw data. Bottom panels emphasize motile mitochondria with red and blue lines transiting left to right or right to left, respectively. (c-f) Quantitative group data demonstrating effect of MiM111 on CMT2A DRG mitochondrial aspect ratio (c), motility (d, e), neuronal process length and branching (e), and proportion of neuronal process termini containing mitochondria (f).

The online version of this article includes the following figure supplement(s) for figure 5:

**Figure supplement 1.** Mitofusin activation with Chimera C reverses mitochondrial pathology and stimulates growth of CMT2A dorsal root ganglion neurons in vitro.

**Figure supplement 2.** Time course studies of DRG mitochondria responses to mitofusin activation after aspiration axotomy.

chambers (*Figure 6a*). Adenoviral Cre was then used to activate the CMT2A MFN2 T105M transgene, followed after 48 hr by aspiration amputation of the branched axonal termini (*Figure 6b and c*). Mitochondrial motility and aspect ratio were measured 1 hr before and after axotomy; axon regrowth was measured 3 days after axotomy. The aspect ratio of mitochondria in the distal linear axons of normal and CMT2A neurons was unaffected either by axotomy or by MiM111 (*Figure 6d*, left panel). By contrast, and consistent with a previous study in normal neurons (*Zhou et al., 2016*), mitochondrial motility was reduced by axotomy (*Figure 6d* middle panels). Mitofusin activation with MiM111 after axotomy restored mitochondrial motility and neuronal outgrowth to pre-axotomy levels. Thus, the link between experimentally activating mitofusins, the subsequent increase in mitochondrial transport, and enhanced neuronal growth/repair was consistent for mouse CMT2A sciatic nerve axons in vivo, cultured CMT2A DRG neuron outgrowth, and cultured CMT2A cortical nerve regrowth after distal axotomy.

## Discussion

These preclinical studies show that activating endogenous normal mitofusins can improve stable neuromuscular dysfunction caused by a CMT2A *MFN2* mutant. Pharmacological mitofusin activation enhanced CMT2A neuron growth in vivo and in vitro by promoting mitochondrial fitness and

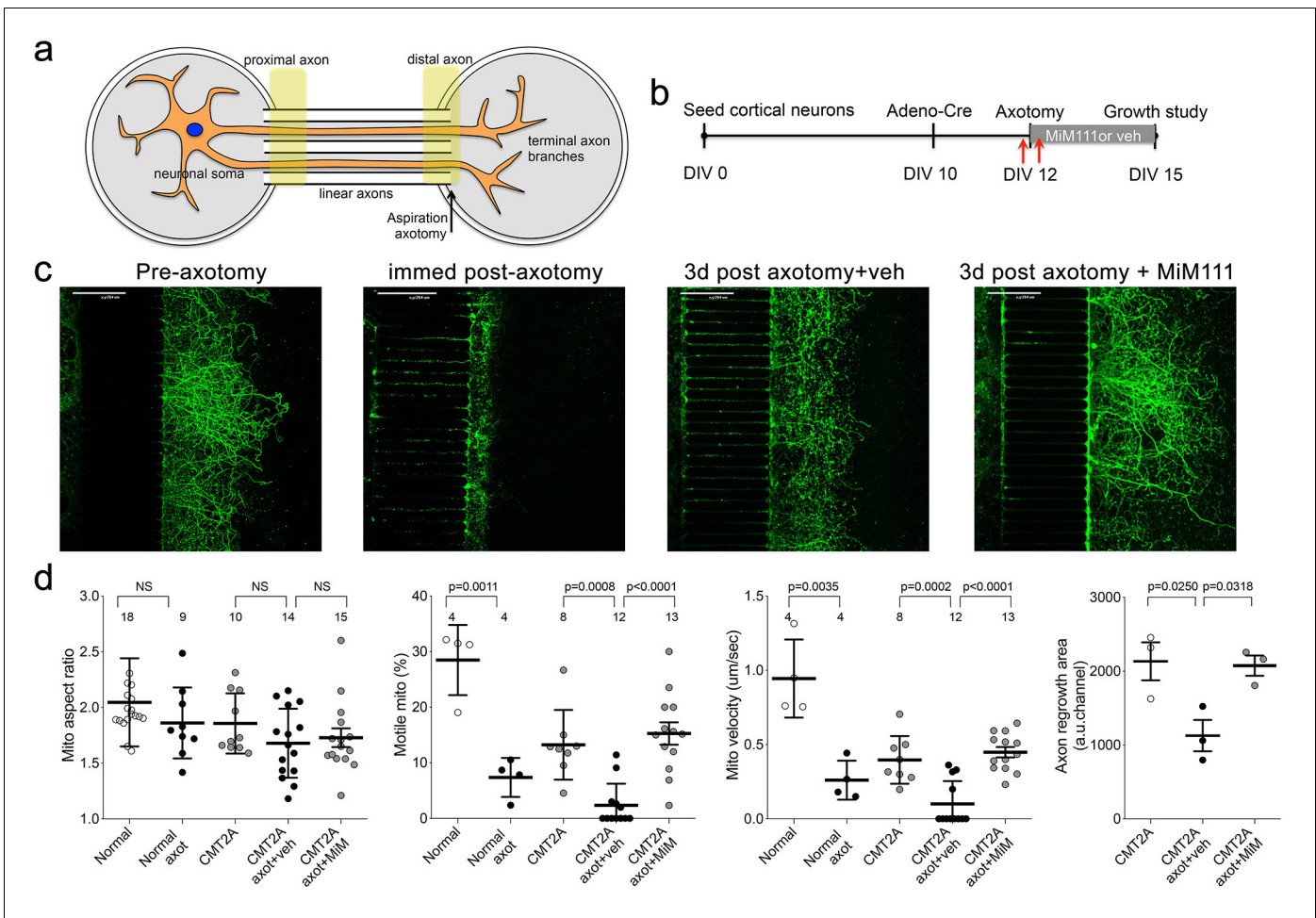

**Figure 6.** Mitofusin activation stimulates post-axotomy regrowth of CMT2A cortical neurons in vitro. (**a**) Schematic depiction of microfluidic platform. Yellow areas show proximal axon where mitochondrial motility was measured and distal axon where mitochondrial aspect ratio was measured. (**b**) Experimental design. DIV is days in vitro. Red arrows are times of pre- and immediate post-axotomy mitochondrial studies. (**c**) Representative images of CMT2A neuron terminal branches at different times relative to aspiration axotomy. (**d**) Quantitative group data demonstrating effect of MiM111 on CMT2A cortical neuron mitochondrial aspect ratio (left panel), motility (middle panels), and axon length (right panel).

transport, thereby reversing CMT2A-associated mitochondrial fragmentation, depolarization and stasis. We believe the key benefit that accrued from directly activating mitofusin-mediated mitochondrial fusion and motility was improved delivery of healthy mitochondria to neuromuscular junctions and axon growth buds. To our knowledge, this is the first report of any experimental intervention that fully reverses in vivo CMT2A phenotypes, demonstrating the feasibility of a clinically translatable disease-modifying therapeutic modality for this incurable condition.

Our studies integrated findings from multiple complementary experimental platforms. Motor neurons directly reprogrammed from CMT2A patient fibroblasts have not previously been described, and provided a platform in which effects of therapeutic interventions could be assessed on different patients' mutations and individual genetic backgrounds using a disease-relevant cell type. Compared to iPS cell-derived CMT2A neurons (*Rizzo et al., 2016*; *Saporta et al., 2015*), direct reprogramming more faithfully reproduced prototypical CMT2A mitochondrial phenotypes. Compared to the parental patient fibroblasts (Dang et al., in preparation), neurons permitted assessment of the CMT2A-associated mitochondrial motility disorders. While CMT2A has both sensory and motor neuropathy components, we reprogrammed specifically for motor neurons because motor components of this disease are the major source of patient disability.

All mouse disease models have advantages and limitations. Our CMT2A mouse model expresses human *MFN2* T105M using a 'motor neuron selective' promoter (*vide infra*), and therefore does not exhibit sensory nerve involvement that is sometime manifested in clinical CMT2A. However, compared to other CMT2A mice, the current model more faithfully recapitulates CMT2A neuromuscular dysfunction that is the dominant cause of morbidity in the human condition (*Fridman et al., 2015*; *Feely et al., 2011*; *Bombelli et al., 2014*; *Yaron and Schuldiner, 2016*; *Berciano et al., 2017*). We previously used young adult mice carrying this combination of MFN2T105M and *HB9*-Cre alleles to evaluate effects of a topically applied prototype mitofusin activator on mitochondrial motility in sciatic nerve axons ex vivo (*Rocha et al., 2018*). It was not known if, with age, these mice would develop neuromuscular signs similar to clinical CMT2A. As shown here, motor function in these mice is normal at age 10 weeks, but declines until age 30 weeks whereupon it stabilized for at least another 20 weeks. This pattern is similar to the clinical course of CMT2A, in which apparently normal children typically manifest neuromuscular signs in the mid first decade of life, exhibit progressive loss of motor function in distal extremities over the next 10–15 years, and then stabilize. Moreover, the functional (neuroelectrophysiological), histological, and ultrastructural features of axonal tissue in the mice were similar to the human condition. Together with the positive response to mitofusin activation in patient neurons, the improvement in neuromuscular function and cell/organelle pathology in MiM111-treated CMT2A mice supports the approach of mitofusin activation for the clinical disease.

Perhaps the most remarkable finding here is that mitofusin activation reversed the signs of CMT2A in mice with severe, stable disease. Every measured endpoint was improved, including gross neuromuscular function (RotaRod), electrophysiological metrics of neuromuscular integrity (CMAP), read-outs for axon degeneration, and multiple histological and ultrastructural assays of mitochondrial pathology in neurons. In vivo and in vitro results pointed to enhanced CMT2A neuron repair and regrowth as a central reason for phenotype reversal. Because it is not possible to functionally dissociate mitofusin-mediated increases in mitochondrial fusion and motility, it is unclear if one or the other of these responses preferentially underlies the neuroregenerative effects of mitofusin activation. However, it seems reasonable to postulate that mitochondrial delivery to distal neurons has greatest importance in the long peripheral nerves innervating hands, forearms, feet, and forelegs, i.e. those areas most impacted in CMT2A (*Fridman et al., 2015*; *Feely et al., 2011*; *Bombelli et al., 2014*; *Yaron and Schuldiner, 2016*; *Berciano et al., 2017*). In agreement with this notion, we observed a positive correlation between mitochondrial delivery to or occupancy of axonal termini and CMT2A neuron growth in vivo and in vitro.

As introduced above, damaging *MFN2* mutations are a straightforward cause of CMT2A, but *MFN2* multifunctionality complicates delineating the underlying cellular pathology (*Filadi et al., 2018*; *Dorn, 2020*). For this reason, the specific functional benefits accruing from mitofusin activation in CMT2A cannot unambiguously be defined. Mitochondrial fusion and motility are impaired in CMT2A (*Chen and Chan, 2006*) and allosteric mitofusin activation corrects both of these parameters (*Franco et al., 2016*; *Rocha et al., 2018*; *Dang et al., 2020*). Mitochondrial respiratory dysfunction, measured here as loss of inner membrane polarization, is a consequence of diminished fusion-

mediated homeostatic repair (*Chen and Chan, 2006*), and its improvement can therefore also be explained by enhanced fusogenicity. *MFN2* has a role in mitophagic mitochondrial quality and quantity control (*Chen and Dorn, 2013*; *Gong et al., 2015*), and allosteric mitofusin activation suppressed the increase in autophagy/mitophagy induced by CMT2A mutant MFN2 T105M in cultured cells (*Rocha et al., 2018*). Finally, MFN2 can mediate physical interactions and calcium signaling between mitochondria and endoplasmic reticulum that may also have a role in CMT2A (*Larrea et al., 2019*), but effects of mitofusin activation on mitochondria-reticular interactions have not been described.

Here, we studied two structurally distinct but functionally similar allosteric mitofusin activators, a new class of drug that is the first to directly enhance mitochondrial fusion and transport. Although the prototype compounds were found not to be 'druggable', a new generation of mitofusin activators have addressed pharmaceutical limitations of the initial chemical series (*Dang et al., 2020*). As described previously (*Rocha et al., 2018*; *Dang et al., 2020*), mitofusin activators have minimal effects in normal cells, likely because increasing the probability that mitofusins are in their open/ 'active' conformation is a subtle intervention that can be readily compensated for in the absence of a pre-existing imbalance between mitochondrial fusion and fission. The current in vivo studies used a short-acting compound administered once daily to evaluate the effects of intermittent, or burst, mitofusin activation on CMT2A neuromuscular dysfunction. We considered that continuous long-term activation of mitochondrial fusion and transport might possibly be deleterious (*El Fissi et al., 2018*) (although it is worth noting that adverse effects of MFN1 and MFN2 overexpression in transgenic mice have not been reported). Moreover, we reasoned that the problem underlying CMT2A is the cumulative effects of long-term mitochondrial stasis and dysfunction on mitochondrial fitness and neuromuscular integrity. This scenario can explain why CMT2A progresses over many years in people and many weeks in mice. Our aim with burst activation was to turn back the disease clock through daily re-setting of mitochondrial function. By intermittently mobilizing healthy mitochondria to distal areas of physiological need, and simultaneously removing senescent or impaired mitochondria, neuron repair, renewal, and neuromuscular signaling were improved.

A mouse is not a man and human neuroregenerative capacity declines with age (*Mattson and Magnus, 2006*). For this reason we do not expect that mitofusin activation can fully reverse CMT2A phenotypes in older human patients with long-term stable disease. Nevertheless, the current results suggest that pharmacological mitofusin activation could offer the first disease-altering therapy for younger CMT2A patients. An intriguing possibility is that mitofusin activation may also have a therapeutic role in some of the many other neurodegenerative conditions not directly caused by mitofusin defects wherein mitochondrial fusion or transport are defective (Dang et al., in preparation; *Knott et al., 2008*; *Burté et al., 2015*).

## Materials and methods

Key resources table

| Reagent type (species) or resource | Designation | Source or reference | Identifiers | Additional information |
|---|---|---|---|---|
| Gene *Mus musculus* | Mfn-2 | NCBI Gene | Gene ID: 170731 | **MFN2** ENSMUSG00 000029020 |
| Gene (Human) | *MFN-2* | NCBI Gene | Gene ID: 9927 | MFN2 ENSG0000 0116688 |
| Genetic reagent (*M. musculus*) | Rosa-STOP-mMFN Thr105Met (T105M) mice | (C57BL/6 Gt(ROSA) 26 Sortm1 (CAG MFN2*T105M) Dple/) | The Jackson Laboratory: 025322 | C57Bl/6 |
| Genetic reagent *M. musculus* | HB9-Cre mice | (B6.129S1-Mnx1tm4 (cre)Tmj/J) | The Jackson Laboratory : 006600 | C57Bl/6 |

*Continued on next page*

Continued

| Reagent type (species) or resource | Designation | Source or reference | Identifiers | Additional information |
|---|---|---|---|---|
| Genetic reagent *M. musculus* | C57BL/6J mice | C57Bl/6 | The Jackson Laboratory : 000664 | C57Bl/6 |
| *Mfn2 null M. musculus* | Mfn2 null MEFs | ATCC | CRL-2994 | Murine embryonic fibroblasts |
| *Mfn1/Mfn2 null M. musculus* | Mfn1 and Mfn2 double knock out MEFs | ATCC | CRL-2993 | Murine embryonic fibroblasts |
| *Mfn1 null M. musculus* | Mfn1 null MEFs | ATCC | CRL-2992 | Murine embryonic fibroblasts |
| Cell line (*H. sapiens*) | Dermal fibroblast (MFN2 T105M) | Dr. Robert H. Baloh (Cedars Sinai) | | Female |
| Cell line (*H. sapiens*) | Dermal fibroblast (MFN2 H361Y) | Dr. Robert H. Baloh (Cedars Sinai) | | Male |
| Cell line (*H. sapiens*) | Dermal fibroblast (MFN2 R274W) | Dr. Barbara Zablocka (Mossakowski Med Res Ctr) | PMID:28076385 | Male |
| Cell line (*H. sapiens*) | Dermal fibroblast (MFN2 R364W) | Dr. Michael E. Shy (University of Iowa) | | Female |
| Cell line (*H. sapiens*) | Dermal fibroblast (Normal) | NINDS | ND34769 | Female |
| Cell line (*H. sapiens*) | Dermal fibroblast (Normal) | NINDS | ND36320 | Female |
| Cell line (*H. sapiens*) | Dermal fibroblast (Normal) | NINDS | ND29510 | Female |
| Transfected construct (Human Adenovirus Type5 (dE1/E3)) | Adenovirus β-galactosidase | Vector Biolabs | Cat#: 1080 | |
| Transfected construct (Human Adenovirus Type5 (dE1/E3)) | Adenovirus Mito-Ds-Red2 | Signagen | Cat#: 12259 | |
| Transfected construct (Human Adenovirus Type5 (dE1/E3)) | Adenovirus Cre-recombinase | Vector Biolabs | Cat#: 1794 | |
| Recombinant DNA reagent | rtTA-N144 (plasmid) | Addgene | Cat#: 66810 | Lentiviral construct to transfect and express the plasmid |
| Recombinant DNA reagent | pTight-9-124-BclxL (plasmid) | Addgene | Cat#: 60857 | Lentiviral construct to transfect and express the plasmid |
| Recombinant DNA reagent | LHX3-N174 and ISL1-N174 (plasmid) | PMID:28886366 | | Lentiviral construct to transfect and express the plasmid |

*Continued on next page*

Continued

| Reagent type (species) or resource | Designation | Source or reference | Identifiers | Additional information |
|---|---|---|---|---|
| Antibody | Anti-Mfn-2 (Mouse monoclonal) | AbCAM | Cat#: ab56889 | (1:1000) |
| Antibody | Anti-COX-IV (Rabbit polyclonal) | AbCAM | Cat#: ab16056 | (1:1000) |
| Antibody | Anti-Stathmin-2 (Rabbit polyclonal) | Novus Biologicals | Cat#: NBP1-49461 | (1:1000) |
| Antibody | Anti-GAPDH (Mouse monoclonal) | AbCAM | Cat#: ab8245 | (1:3000) |
| Antibody | Anti-FSP-1 (Rabbit polyclonal) | Novus Biologicals | Cat#: NBP1-49461 | (1:400) |
| Antibody | Anti-MNX1 (Mouse monoclonal) | DSHB | Cat#: 81.5C10 | (2 µg/ml) |
| Antibody | Anti-β-tubulin III (Mouse monoclonal) | Biolegend | Cat#: 801201 | (1:200) |
| Antibody | Alexa-Fluor 488 (Goat anti-mouse) | ThermoFisher | Cat#: A11029 | (1:400) |
| Antibody | Alexa- Fluor 488 (Goat anti-rabbit) | ThermoFishe | Cat#: A11008 | (1:400) |
| Antibody | (Goat anti-rabbit IgG) | ThermoFisher | Cat#: 31460 | (1:3000) |
| Antibody | Alexa- Fluor 594 (Goat anti rabbit) | ThermoFisher | Cat#: A32740 | (1:400) |
| Antibody | (Peroxidase-conjugated anti-mouse IgG) | Cell Signaling | Cat#: 7076S | (1:3000) |
| Antibody | (α-Bungarotoxin Alexa flour 594) | ThermoFisher | Cat#: B12423 | (0.5 µg/ml) |
| Sequence-based reagent | HB9CRE Fw | The Jackson Laboratory | 006600 | CTAGGCCACAGA ATTGAAAGATCT |
| Sequence-based reagent | HB9CRE Rv | The Jackson Laboratory | 006600 | GTAGGTGGAAA TTCTAGCATCATCC |
| Sequence-based reagent | HB9CRE TG Fw | The Jackson Laboratory | 006600 | GCGGTCTGGCA GTAAAAACTATC |
| Sequence-based reagent | HB9CRE TG Rv | The Jackson Laboratory | 006600 | GTGAAACAGCAT TGCTGTCACTT |
| Sequence-based reagent | Mfn2 T105M M Fw | The Jackson Laboratory | 025322 | GACCCCGTT ACCACAGAAGA |
| Sequence-based reagent | Mfn2 T105M M Rv | The Jackson Laboratory | 025322 | AACTTTGTCC CAGAGCATGG |

Continued

| Reagent type (species) or resource | Designation | Source or reference | Identifiers | Additional information |
|---|---|---|---|---|
| Sequence-based reagent | Mfn2 T105M Wt Fw | The Jackson Laboratory | 025322 | AAGGGAGCTGC AGTGGAGTA |
| Sequence-based reagent | Mfn2 T105M Wt Rv | The Jackson Laboratory | 025322 | CCGAAAATCT GTGGGAAGTC |
| Sequence-based reagent | MFN2 T105M Fw | This paper | PCR primers for cell line mutation validation | TTGCACTGAA TAGGGCTTTG |
| Sequence-based reagent | MFN2 T105M Rv | This paper | PCR primers for cell line mutation validation | CATTCACCTC CACAGGGTG |
| Sequence-based reagent | MFN2 R274W Fw | This paper | PCR primers for cell line mutation validation | CGTGGTAGGTG TCTACAAGAAGC |
| Sequence-based reagent | MFN2 R274W Rv | This paper | PCR primers for cell line mutation validation | CTGGTGAGG GCTGATGAAAT |
| Sequence-based reagent | MFN2 H361Y and R364W Fw | This paper | PCR primers for cell line mutation validation | CCTGGCAGTGA AAACCAGAG |
| Sequence-based reagent | MFN2 H361Y and R364W Rv | This paper | PCR primers for cell line mutation validation | AAGGCGTGT CCTAACTGCC |
| Chemical compound, drug | Trans-MiM111 | Mitochondria in Motion, Inc | Cpd 13b in PMID:32506913 | |
| Chemical compound, drug | Chimera C | Paraza Pharma | Cpd 2 in PMID:32506913 | |
| Chemical compound, drug | Papain | Sigma | Cat#: P4762 | |
| Chemical compound, drug | Laminin | Sigma | Cat#: L2020 | |
| Chemical compound, drug | Poly-d-Lysine | Sigma | Cat#: P7886 | |
| Chemical compound, drug | Poly-ornithine | Sigma-Aldrich | Cat#: P4957 | |
| Chemical compound, drug | Fibronectin | Sigma-Aldrich | Cat#: F4759 | |
| Chemical compound, drug | Polybrene | Sigma-Aldrich | Cat#: H9268 | |
| Chemical compound, drug | Doxycycline | Sigma-Aldrich | Cat#: D9891 | |
| Chemical compound, drug | G418/Geneticin | Invitrogen | Cat#: 10131-035 | |

Continued

| Reagent type (species) or resource | Designation | Source or reference | Identifiers | Additional information |
|---|---|---|---|---|
| Chemical compound, drug | Retinoic Acid | Sigma | Cat#: R2625 | |
| Chemical compound, drug | BDNF, NT-3, CNTF, GDNF | Peprotech | Cat#: 450-02, Cat#: 450-03, Cat#: 450-13, Cat#: 450-10 | |
| Chemical compound, drug | Dibutyryl cAMP | Sigma | Cat#: D0627 | |
| Chemical compound, drug | Valproic acid | Sigma | Cat#: 676380 | |
| Chemical compound, drug | Puromycin | Invitrogen | Cat#: A11138-03 | |
| Chemical compound, drug | Collagenase | Worthington Biochemical | Cat#: 41J12861 | |
| Chemical compound, drug | (2-Hydroxypropyl)-$\beta$-cyclodextrin | Sigma | Cat#: 332607 | |
| Chemical compound, drug | Carbonyl cyanide-$p$-trifluoro methoxyphenyl hydrazone | Sigma | Cat#: C2759 | |
| Chemical compound, drug | B27 supplement | Gibco | Cat#: 17504-044 | |
| Chemical compound, drug | Insulin-transferrin-sodium selenite | Sigma | Cat#: 1884 | |
| Chemical compound, drug | Glucose | Sigma | Cat#: G5767 | |
| Chemical compound, drug | L-glutamine | Gibco | Cat#: 25030-149 | |
| Chemical compound, drug | Goat serum | Jackson Immunoresearch | Cat#: 005-000121 | |
| Chemical compound, drug | Glutaraldehyde | Electron Microscopy Science | Cat#: 16216 | |
| Chemical compound, drug | MitoTracker Green | Thermo Fisher | Cat#: M7514 | |
| Chemical compound, drug | Calcein AM | Thermo Fisher | Cat#: C3100MP | |
| Chemical compound, drug | Hoechst | Thermo Fisher | Cat#: H3570 | |
| Chemical compound, drug | MitoTracker Orange | Thermo Fisher | Cat#: M7510 | |

*Continued*

| Reagent type (species) or resource | Designation | Source or reference | Identifiers | |
|---|---|---|---|---|
| Chemical compound, drug | Tetramethylrhodamine ethyl ester | Thermo Fisher | Cat#: T-669 | |
| Software, algorithm | ImageJ | C. A. Schneider | https://imagej.net/Sholl_Analysis | |
| Software, algorithm | Viasys Healthcare Nicolet Biomedical instrument with Viking Quest version 11.2 software | Middleton | Cat#: OL060954 | **Additional information** |
| Software, algorithm | Gallios instrument with FlowJo 10 software | Beckman Coulter | N/A | |
| Other | RotaRod | Ugo Basile | Cat#: 47650 | |
| Other | XonaChips | Xona Microfluidics | Cat#: XC450 | |

## Mouse lines

Rosa-STOP-mMFN Thr105Met (T105M) mice (C57BL/6 Gt(ROSA)26 Sortm1 (CAG-*MFN2*\*T105M) Dple/J) from The Jackson Laboratory (Bar Harbor, Maine, USA; Stock No: 025322, RRID:MGI:_JAX:025322) were crossed to HB9-Cre mice (B6.129S1-Mnx1tm4(cre)Tmj/J) from The Jackson Laboratory (Stock No: 006600, (RRID:MGI:_JAX:006600)) to generate neuron-targeted MFN2 T105M mice. HB9 is a motoneuron-specific promoter (*Yang et al., 2001*), but JAX data indicates that this HB9-Cre line also drives expression in some sensory DRG neurons (https://images.jax.org/webclient/img_detail/20564/). All experimental procedures were approved by Washington University in St. Louis School of Medicine Animal Studies Committee; IACUC protocol number 19–0910, Exp:12/16/2022.

## Cell lines

Normal mouse embryonic fibroblasts (MEFs) were prepared by enzymatic dissociation from embryonic day E.13.5–14.5 C57BL/6J mice (The Jackson Laboratory Cat:# 000664, RRID:IMSR_JAX:000664). *Mfn1* null and *Mfn2* null *Mfn1/Mfn2* double null MEFs fibroblasts were purchased from American Type Culture Collection (ATCC Manassas, Virginia, USA) (CRL-2992, RRID:CVCL_L691 and CRL-2994, RRID:CVCL_L692 and CRL-2993, RRID:CVCL_L693 respectively). Human fibroblast: Dermal fibroblast (MFN2 T105M) and Dermal fibroblast (MFN2 H361Y) from Dr. Robert H. Baloh (Cedars Sinai), Dermal fibroblast (MFN2 R274W) from Dr. Barbara Zablocka (Mossakowski Med Res Ctr), Dermal fibroblast (MFN2 R364W) from Dr. Michael E. Shy (University of Iowa). Dermal fibroblast (Normal) from NINDS respectively: ND34769, (RRID:CVCL_EZ04, ND36320, RRID:CVCL_EZ16 and ND29510, RRID:CVCL_Y813).

## Viral vectors

Adenovirus expressing human FLAG-hMFN2 -T105M was prepared at Vector Biolabs (Malvern, PA, USA). Adenoviri expressing β-galactosidase (Ad-CMV-β-Gal; #1080), and Ad-Cre (#1794) were purchased from Vector Biolabs. Adenovirus for Mito-Ds-Red2 came from Signagen (Cat:#SL1007744). Lentivirus packaging vectors: psPAX2 (Addgene, Cat#: 12260) pMD2.G (Addgene, Cat#: 12259), Lentiviral vectors with recombinant DNA: rtTA-N144 (Addgene, Cat#: 66810) pTight-9–124-BclxL (Addgene, Cat#: 60857), human LHX3-N174 and human ISL1-N174 were packaged and used as described (*Abernathy et al., 2017*).

## Antibodies

Mouse monoclonal anti-mitofusin 2 (Cat # ab56889 - 1:1000, RRID:AB_2142629), anti-COX-IV (Cat #ab16056 - 1:1000, RRID:AB_443304) and anti-GAPDH (Cat #ab8245 - 1:1000, RRID:AB_2107448)

were from AbCAM (Cambridge, MA, USA). Rabbit polyclonal anti-Stathmin-2/Superior Cervical Ganglion 10 (SCGN10; Cat # NBP1-4946, RRID:AB_10011569) was from Novus Biologicals (Littleton, CO, USA).Rabbit polyclonal FSP-1 was from Sigma Millipore (Cat # 07–2274, RRID:AB_10807552). Anti-mouse monoclonal -MNX1was from DSHB (1:10, Cat# 81.5C10, RRID:AB_2145209). Mouse monoclonal anti-β -tubulin III (Cat # 801201- 1:500, RRID:AB_2313773) was from Biolegend (San Diego, CA, USA). Peroxidase-conjugated anti-mouse IgG (Cat #7076S - 1:1000, RRID:AB_330924) was from Cell Signaling (Danvers, MA, USA). Goat anti-rabbit IgG (Spicier reactivity Goat, Host/Isotype Rabbit/IgG; Cat #31460, RRID:AB_228341) and Alexa-Fluor 488 anti-mouse ThermoFisher (Waltham, MA, USA Cat #A11008, RRID:AB_143165 ). α-Bugarotoxin Alexa flour 594 was from ThermoFisher, Waltham, MA, USA Cat:# B12423.

## PCR genotyping of MFN2 mutations in CMT2A patient fibroblasts

DNA was extracted from $5 \times 10^6$ primary human fibroblasts using the DNeasy blood and tissue kit (Qiagen, Cat#: 69506) according to manufacturer's protocol. PCR used Taq Plus Master Mix 2X (Cat#: BETAQR-L, Bulls eye). 50 ng of genomic DNA template, and the following primers:

> (*MFN2* T105M) - 5'- TTGCACTGAATAGGGCTTTG- 3'
> 5'- CATTCACCTCCACAGGGTG- 3'
> (*MFN2* R274W) - 5'- CGTGGTAGGTGTCTACAAGAAGC- 3'
> 5'- CTGGTGAGGGCTGATGAAAT- 3'
> (*MFN2* H361Y, R364W) - 5'-CCTGGCAGTGAAAACCAGAG- 3'
> 5'- AAGGCGTGTCCTAACTGCC- 3'.

PCR products were purified using PureLink Quick Gel Extraction Kit (Cat#: K21000-12, Invitrogen). Sanger sequencing of PCR products was performed at GENEWIZ.

## Cultured cells

Direct reprogramming of human motor neurons from patient fibroblasts used the procedure as described (*Abernathy et al., 2017*). Reprogramming cocktail consisted of 1 ml concentrated lentivirus containing the reverse tetracycline-controlled transactivator (rtTA; Addgene, Cat# 66810), 1 ml virus containing pT-BclXL-9/9*−124, 125 µl virus containing motor neuron transcription factor ISL1, and 125 µl virus containing motor neuron transcription factor LHX3 with polybrene (8 µg/ml; Sigma-Aldrich, Cat# H9268). Human skin fibroblasts of low passage number (P4-P7) were spinfected at 37° C for 30 min at 1,000 × G. Doxycycline (Dox, 1 µg/ml; Sigma Aldrich, Cat# D9891) and antibiotics for respective vectors (Puromycin, 3 µg/ml; Invitrogen, Cat# A11138-03; Geneticin, 400 µg/ml; Invitrogen, Cat# 10131–035) were added to culture medium for 4 days after viral transduction. On day 5 cells were re-plated on poly-ornithine/laminin/fibronectin (Sigma, Cat# P4957, # L2020, # F4759) coated 18 mm glass coverslips and on the following day changed to neuronal medium supplemented with Dox (1 µg/ml added every other day), valproic acid (1 mM; Sigma, Cat# 676380), dibutyryl cAMP (200 µM; Sigma, Cat# D0627), BDNF, NT-3, CNTF, GDNF (all 10 ng/ml, Peprotech, Cat# 450–02, #450–03, #450–13, #450–10), retinoic Acid (1 µM; Sigma, Cat# R2625) and antibiotics. Neuronal medium was refreshed by replacing half every 4 days. Antibiotics were discontinued on day 14; Dox was discontinued on day 30. Cells underwent studies beginning on day 35. Motor neurons were identified after formalin fixation by labeling with mouse anti-MNX1 (1:10; DSHB, Cat# 81.5C10) and mouse anti-TUBB3B (1:2000; Biolegend, Cat#PRB-435P-100). Fibroblasts were identified by labeling with rabbit anti FSP-1 (1:200; Sigma, Cat: # 07–2274).

Adult mouse dorsal root ganglion (DRG) neurons were prepared from ~8-week-old HB9Cre-MFN2 Thr105Met flox-stop transgenic mice as previously described (*Rocha et al., 2018*). To comprehensively induce MFN2 T105M transgene expression, the DRGs were infected with Adeno-Cre (M.O.I. of 50) 48 hr prior to study. DRG neurons were distinguished from non-neuronal cells by staining with anti-β-III tubulin.

Mouse cortical neurons were isolated from individual embryonic day E.18.5 MFN2 Thr105Met flox-stop transgenic mice by papain digestion and mechanical dispersion using a published procedure (*Sobieski et al., 2015*). Briefly, mouse brain cortices were isolated under a dissecting microscope and sliced into 0.5–1 mm thick sections in Leibovitz's L-15 Medium (Gibco Cat:#11415–064) containing BSA (0.23 mg/ml, Sigma Cat:#A7030). Papain (1 mg/ml, Sigma Cat #P4762) was added and the tissue digested for 20 min at 37°C. The papain solution was replaced and micropipettes

used to triturate the solution until no more tissue was visible. Cortical neurons were plated in microfluidic neuron XonaChip chambers as described below.

## Imaging

Static confocal imaging of cultured neurons used triple-stained with MitoTracker Green (200 nM; Invitrogen, Thermo Fisher Scientific Cat:# M7514) to visualize mitochondria, tetramethylrhodamine ethyl ester (TMRE, 200 nM, Invitrogen Thermo Fisher Scientific Cat:# T-669) that labels mitochondria with normal polarization of the mitochondrial inner membrane, and Hoechst (10 µg/ml; Invitrogen, Thermo Fisher Scientific Cat:# H3570) that stains nuclei blue as described (*Franco et al., 2016*). Static live cell images were acquired on a Nikon Ti Confocal microscope using either a 60 × 1.3 NA oil-immersion objective or 10 × 0.3 NA dry objective, in Krebs-Henseleit buffer (138 NaCl, 3.7 nM KCL, 1.2 n M KH2PO4, 15 nM glucose, 20 nM HEPES pH: 7.2–7.5, and 1 mM CaCl$_2$): laser excitation was 488 nm with emission at 510 nm for MitoTracker Green and Ad-Mito GFP, 549 nm with emission at 590 nm for TMRE, and 306 nm with emission 405 nm for Hoecsht and DAPI.

Axon branching analysis of CMT2A mouse DRGs was performed at various times after isolation and plating, as indicated. In some studies neurons were infected with Ad-mito-RFP to label mitochondria red. Cells were fixed and labeled with anti-β-tubulin III (1:200 in 10% goat serum in PBS) to identify neurons. Images were acquired using the 10x objective and excitation at 488 nm/emission 510 nm for Alexa-Flour 488 and 579 excitation/599 emission for mito-RFP. Sholl analysis of axonal branching used ImageJ (*Schneider et al., 2012*) and an open source plugin (https://imagej.net/Sholl_Analysis). Briefly, a starting radius was set to encompass the soma of β-tubulin III-positive DRG neurons and concentric circles established at 10 micron increments, to 40 microns. Numbers of axon and radii intersections were totaled for all circles to derive intersection number, which is a measure of axonal branching. Special attention was given to ensure that there was uniform staining along all parts of the DRG soma and axons so that the plugin was able to accurately assess the number of intersections accurately.

Video confocal studies of mitochondrial motility studies in neurons and sciatic nerves used time-lapse imaging (1 frame every 5 s) for 121 frames (10 min, sciatic nerve) or 180 frames (15 min, cultured neurons) at 37°C on a Nikon A1Rsi Confocal Microscope using a 40x oil objective as described (*Rocha et al., 2018*). Cultured neuron mitochondria were labeled with Adeno-mitoDsRed2 or Mito-Tracker Orange (200 nM, Invitrogen Thermo Fisher Scientific Cat:# M7510) excited at 561 nm, emission 585 nm. Sciatic nerve axon mitochondria were labeled with TMRE. Kymographs and quantitative data were generated using an Image-J plug-in.

In vitro microfluidic studies of axon growth used primary cortical neurons isolated from embryonic day 18.5 MFN2 T105M flox-stop mice. 50,000–90,000 suspended cells in 20 µl of Earle's Minimal Essential Medium (MEM; #11090–081; Gibco) supplemented with 5% FBS (Gibco #16140–063), 5% horse serum (HS) (Gibco #26050–070), 400 µM L-glutamine (Gibco #25030–149), 50 units/ml each penicillin/streptomycin (Gibco #15070–063) and 0.3% glucose (Sigma G 5767) (5–5 media) was added to the left chambers of XonaChips with 450 µm microgroove barriers (#XC450; Xona Microfluidics, Temecula, CA, USA) coated with 0.5 mg/ml poly(D)lysine (Sigma #P7280). Ten minutes thereafter, 150 µl of 5–5 medium supplemented with 0.5 µl insulin-transferrin-sodium selenite (Sigma I 1884) was added to each well and neurons cultured under standard conditions. After 24 hr the medium was changed to neurobasal medium (#21103–049; Gibco, Carlsbad, CA, USA), 1x B27 supplement (#17504–044, Gibco, Carlsbad, CA, USA), 50 units/ml each penicillin/streptomycin (#15070–063; Gibco, Carlsbad, CA, USA) and 400 µM L-glutamine (#25030–149; Gibco, Carlsbad, CA, USA) with feeding every 2 to 3 days until axotomy (DIV 12), and infected with adeno-Cre for 48 hr to induce MFN2 Thr105Met expression. Vacuum aspiration axotomy and post-axotomy regrowth analyses were performed as described (*Zhou et al., 2016*). Aspiration axotomy was followed by application of fresh neuron feeding medium containing MIM111 (100 nM) or its vehicle (Me$_2$SO, 1:1,000). Cells were fixed in situ; axonal outgrowth and post-axotomy regrowth were analyzed by confocal analysis of β-III tubulin positive cells. The area of βIII-tubulin signals above the same threshold within a 1024 × 1024 image that covers all axon segments extending from microgrooves was measured using ImageJ (NIH) and reported as pixels density of axon segments extending from an average of all microgrooves.

Immunoblot analysis was performed on mouse sciatic nerve proteins size-separated on 10% SDS-PAGE gels (Biorad Cat# 456–1036) and transferred to 0.45 µM Polyvinylidene fluoride (PVDF)

membranes (GE- Amersham, Freiburg, Germany Cat# 10600023). Membranes were blocked with 5% non-fat milk for 30 min and incubated with primary antibody overnight at 4°C. Peroxidase-conjugated secondary antibodies and Chemiluminescence Substrate (Thermo Scientific #32132) were used for signal detection. Quantification of immunoreactive proteins was performed on a LI-COR Odyssey infrared detection system (Lincoln, NE, USA, version 1.0.17).

## Flow cytometric analysis of mitochondrial electrochemical potential

Cultured neurons were stained in situ with TMRE (200 nM, Invitrogen Thermo Fisher Scientific Cat:# T-669) for 30 min at 37°C in 5% $CO_2$–95% air, washed twice in PBS, and released from culture substrates with 0.05% Trypsin-EDTA (Gibco, cat:# 1995647). After centrifugation, the DRG pellets were re-suspended in 200 µl of FACS buffer (PBS 1X, BSA 1X, 2 Mm EDTA). Flow cytometry of TMRE fluorescence was performed on a Gallios instrument (Beckman Coulter) and analyzed using FlowJo10 software. ~3500 events were acquired for each sample. Data are presented as Mean Fluorescence Intensity per experiment. In some studies, carbonyl cyanide-p-trifluoromethoxyphenylhydrazone (FCCP, 10 µM for 1 hr) (Sigma, Cat #C2759) was added as a positive control for mitochondrial depolarization.

## Evaluation of neuromuscular phenotypes in CMT2A mice

Rotarod studies were performed on mice initially acclimated to the RotaRod (Ugo Basile, Gemonio, VA, Italy;# 47650) at a speed of 5 r.p.m. CMT2A mice underwent RotaRod evaluations weekly from 10 to 50 weeks for disease development, and 4 and 8 weeks after mitofusin activator therapy. The acceleration protocol increased from 5 to 40 r.p.m over 120 s and then maintained 40 r.p.m. indefinitely. Each mouse underwent five separate trials per testing event with 5 min rest between trials. Latency (time to falling off) was averaged for all trials.

Neuroelectrophysiologic recordings of tibialis/gastrocnemius compound muscle action potentials (CMAP) were performed with a Viasys Healthcare Nicolet Biomedical instrument (Middleton, WI, USA Cat:# OL060954) running Viking Quest version 11.2 software by an operator (A.F.) blinded to genotype and treatment group. Mice were anesthetized with isofluorane (4–5% induction, 1.5% maintenance), shaved, and the proximal sciatic nerves stimulated using a needle electrode (Natus, Mundelein, IL, USA Cat:# F-E2-48) with 3.9 mV pulses of 0.002 ms duration. Ring electrodes (Natus, Mundelein, IL, USA Cat:# 291965) were positioned at the mid forelimb at the belly of the tibialis anterior and gastrocnemius muscles to record CMAP. Optimal stimulating electrode position was determined as that giving the greatest CMAP amplitude; 3–4 independent events were recorded and averaged.

## Evaluation of CMT2A mouse responses to MiM111

Twelve 50-week-old CMT2A mice (HB9-Cre + MFN2 Thr105Met flox-stop) and six littermate controls were randomized and blinded to daily intramuscular treatment with MiM111 or vehicle for 8 weeks: under sterile conditions18.75 mg/ml (64.8 mM) MIM 111 was dissolved in 10% $Me_2SO$/90% (2-hydroxypropyl)-β-cyclodextrin (HP-BCD; Sigma, Cat: #332607), sterile-filtered (0.22 µm PVDF, #SLGV033RS, Millipore, Cork, Ireland), and drug- or vehicle-containing syringes were assigned to individual mice by XD using a randomization table. Daily intramuscular injections (biceps femoris muscle, alternating left and right every other day) were performed by AF, who was blinded to both mouse genotype and drug treatment group. Rotarod and neurophysiological testing were performed before, and 4 and 8 weeks after initiation of therapy. Mice were terminated by anesthesia overdose after 8 weeks for tissue studies. Sciatic and mid tibial nerves were dissected from both legs of all mice. For histology and immunohistology the left leg nerves or muscles were fixed in PFA for 2 hr, transferred to 30% sucrose/PBS overnight at 4°C, and embedded in optimal cutting temperature (OCT, Tissue-TEK Cat: 4583) medium for storage at −80°C. Immunostaining with anti-Superior Cervical Ganglion 10 (SCG10) or wheat germ agglutinin labeling (WGA, Cat:#W834, Invitrogen) was performed on 10 µm cryostat sections briefly (5 min) brought to room temperature and then re-cooled to −20°C for 30 min. RGB rightness of the representative images was increased uniformly for presentation purposes.

Mitochondrial occupancy in neuromuscular synaptic junctions was assessed in 10 µm cryostat tibialis muscle sections using anti-COXIV (1:200 in 10% goat serum) to label mitochondria and anti-acetylcholine receptor with α-Bugarotoxin to label neuronal synapses.

Transmission electron microscopy and toluidine blue staining used standard techniques (*Zhou et al., 2019*).

### Data presentation and statistical analyses

Data are reported as means ± SD. Two-group comparisons used Student's t-test; multiple group comparisons used one-way ANOVA, and time-course by treatment group or genotype by treatment group comparisons used two-way ANOVA, with Tukey's post-hoc test for individual statistical comparisons. $p < 0.05$ was considered significant.

Mouse treatment was randomized according to a random integer table (even or odd) and performed by investigators blind to both genotype and treatment status. Post terminal analysis of tissue and cell phenotypes was performed blindly.

Samples size estimation: Using two-sample t-test based on the preliminary data where the coefficients of variation (CV) at 50 weeks were 10% and 15% for rotarod latency and CMAP amplitude, respectively, the study was initially designed to have a sample size of 15 mice/group, providing 80% power at 1-sided $\alpha = 0.05$. Because the therapeutic response for targeted differences was greater than anticipated, the study was completed with a reduced sample size of n = 6/group.

## Acknowledgements

We gratefully acknowledge helpful discussions with Drs. P Needleman, PV Halushka, and D Mochly-Rosen, the specialized assistance of L Zhang., and C Cantoni for flow cytometry analysis. Funding: Supported by NIH R35HL135736, R41NS113642, R41NS115184, Research Grant 628906 from the Muscular Dystrophy Association (GWD), and a McDonnell Center for Cellular and Molecular Neurobiology Postdoctoral Fellowship (AF). GWD is the Philip and Sima K Needleman-endowed Professor and a Scholar-Innovator awardee of the Harrington Discovery Institute, Videoconfocal and electron microscopy studies were performed at the Washington University Center for Cellular Imaging (WUCCI) supported by the Washington University School of Medicine, the Children's Discovery Institute of Washington University (CDI-COREs 2015–515 and 2019–813) and the Foundation for Barnes-Jewish Hospital (3770 and 4672). Competing interests: GWD is an inventor on patent applications PCT/US18/028514 submitted by Washington University and PCT/US19/46356 submitted by Mitochondria Emotion, Inc that cover the use of small molecule mitofusin agonists to treat chronic neurodegenerative diseases, and is a founder of Mitochondria in Motion, Inc, a Saint Louis based biotech R and D company focused on enhancing mitochondrial trafficking and fitness in neurodegenerative diseases. The other authors declare no competing interests. Data and materials availability: All data are available in the manuscript or the supplementary material. Studies using MiM111 were performed under terms of an MTA between Mitochondria in Motion, Inc and Washington University.

## Additional information

### Competing interests

Gerald W Dorn II: GWD is an inventor on patent applications PCT/US18/028514 submitted by Washington University and PCT/US19/46356 submitted by Mitochondria Emotion, Inc that cover the use of small molecule mitofusin agonists to treat chronic neurodegenerative diseases, and is a founder of Mitochondria in Motion, Inc, a Saint Louis based biotech R&D company focused on enhancing mitochondrial trafficking and fitness in neurodegenerative diseases. The other authors declare that no competing interests exist.

### Funding

| Funder | Grant reference number | Author |
|---|---|---|
| National Institutes of Health | R35HL135736 | Gerald W Dorn II |

| National Institutes of Health | R41NS113642 | Gerald W Dorn II |
|---|---|---|
| National Institutes of Health | R41NS115184 | Gerald W Dorn II |
| Muscular Dystrophy Association | 628906 | Gerald W Dorn II |
| McDonnell Center for Cellular and Molecular | Neurobiology Postdoctoral Fellowship | Antonietta Franco |
| Harrington Discovery Institute | Scholar-Innovator awardee | Gerald W Dorn II |

The funders had no role in study design, data collection and interpretation, or the decision to submit the work for publication.

## Author contributions
Antonietta Franco, Resources, Data curation, Funding acquisition, Validation, Investigation, Methodology, Writing - original draft, Writing - review and editing; Xiawei Dang, Resources, Data curation, Formal analysis, Validation, Methodology, Writing - original draft, Writing - review and editing; Emily K Walton, Resources, Methodology; Joshua N Ho, Resources, Validation, Methodology, Special expertise and data interpretation; Barbara Zablocka, Michael E Shy, Resources, Provided essential reagents and intellectual input; Cindy Ly, Validation, Methodology, Provided special expertise and data interpretation; Timothy M Miller, Validation, Methodology, Provided essential reagents and intellectual input; Robert H Baloh, Resources, Validation, Methodology, Provided essential reagents and intellectual input; Andrew S Yoo, Resources, Methodology, Provided essential reagents and intellectual input; Gerald W Dorn II, Conceptualization, Resources, Data curation, Supervision, Funding acquisition, Validation, Investigation, Visualization, Methodology, Writing - original draft, Project administration, Writing - review and editing

## Author ORCIDs
Antonietta Franco http://orcid.org/0000-0002-5487-1800
Xiawei Dang http://orcid.org/0000-0002-0343-7107
Andrew S Yoo http://orcid.org/0000-0002-0304-3247
Gerald W Dorn II https://orcid.org/0000-0002-8995-1624

## Ethics
Animal experimentation: All experimental procedures were approved by Washington University in St. Louis School of Medicine Animal Studies Committee; IACUC protocol number 19-0910, Exp:12/16/2022 (Gerald Dorn, PI).

## Decision letter and Author response
Decision letter https://doi.org/10.7554/eLife.61119.sa1
Author response https://doi.org/10.7554/eLife.61119.sa2

# Additional files

## Supplementary files
• Transparent reporting form

## Data availability
All data generated or analyzed during this study are included in the manuscript.

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
