## [Decision Letter]

**Decision letter after peer review:**

Thank you for submitting your article "Burst mitofusin activation reverses neuromuscular dysfunction in murine CMT2A" for consideration by *eLife*. Your article has been reviewed by two peer reviewers, and the evaluation has been overseen by a Reviewing Editor and Gary Westbrook as the Senior Editor. The following individual involved in review of your submission has agreed to reveal their identity: Delphina Larrea. The reviewers have discussed the reviews with one another and the Reviewing Editor has drafted this decision to help you prepare a revised submission.

Summary:

This work provides preclinical data for MiM111 in CMT2A treatment. Mutations in MFN2 are associated with CMT2A. Mfn2 and Mfn1 are components of the mitochondrial outer membrane fusion machine that form homo and hetero complexes. Mfn2 contains four major structural domains including a GTPase domain, two sequential extended helical bundles (HB1 and HB2) connected by two short loops, and a transmembrane domain. Here the authors describe a new molecule called "MiM111" that in comparison with the previous one described (Chimera C) is metabolically stable with good nervous system bioavailability. These molecules rescue the mitochondrial alterations (mito aspect ratio, depolarization, motility and mito velocity) found in reprogrammed motor neurons from three CMT2A patients expressing different mutations in MFN2. These molecules promote conformational activation of MFN1 and MFN2 by stimulating endogenous mitofusins as reported by the authors previously. The effect of MiM111 was assessed in mouse models of CMT2A in which the integrity and function of Neuromuscular junctions (measured in rotarod) were rescued after MiM111 treatment. Histopathology showed disease reversion including number and size of axons. Finally, the authors show that MiM111 also activates Mitofusins, thereby reversing mitochondrial pathology and stimulating growth of CMT2A DRG neurons in vitro. This work is an extension of two previous papers from this group Rocha et al., 2018 and Dang et al., 2020 and I believe that could benefit CMT2A patients in the future.

Essential revisions:

1) The authors should comment on the additional role of MFN2 as a well-recognized ER-mitochondrial tether at the mitochondrial-associated ER membranes (MAM) (de Brito 2008 Nature, 456, 605., Naon 2016 PNAS., 113, 11249.). MAM is a dynamic platform in which several cellular pathways are regulated (Phospholipids synthesis, Ca2^+^, cholesterol, mito (Pera et al., 2017 EMBO J., 36, 3356-3371). Moreover, a few MAM functions are affected in CMT2A patient fibroblasts (Larrea et al., 2019) and in other sensory neuropathies described in Krols et al., 2019 HMG 28:4). Therefore, authors should demonstrate or at least discuss whether the effects of MiM111 are due to specifically to MFN1 and MFN2 mitochondrial fusion activity or might also be related to other MAM activities.

2) How do the mitofusin activators "activate mitofusins"? In Rocha et al., 2018 author found that mitofusin agonist activity was through stabilization of the fusion-permissive open conformation of endogenous normal MFN1 or MFN2 overcoming the dominant suppression of mitochondrial fusion caused by MFN2 T105M and R94Q. Does MiM111 activate mitofusins in the same way? What is the specificity and selectivity of both activators? Does it affect other mitochondrial GTPase, in addition to MFNs? The authors never evaluated how the supposed target, MFN1/2 (mRNA, protein, or activity) changes. Mitochondrial dynamics and function are a secondary readout and cannot be entirely attributed to MFNs. The current data cannot exclude the possibility that the effect is through other targets rather than MFNs. Could the authors add some controls that show how these activators increase endogenous mitofusin 'activity'? Is it due to increase in endogenous protein stability, transcription/translation or GTPase activity?

3) Regarding the extended used of MiM111 for all CMT2A mutations; could the authors address the fact that studies in the nervous system, fibroblast and reprogrammed motor neurons from CMT2A patients, have displayed a variety of altered mitochondrial morphology, including swelling, degeneration and altered distribution of mitochondria (Verhoeven et al., 2006, Amiott et al., 2008; Larrea et al., 2019; Saporta, et al., 2015). However mitochondrial fragmentation was not observed. Nevertheless, all mutations caused CMT2A. Would MiM111 still be a rational therapy in all these cases? Why does the compound have no effect on control cells? As mentioned above, it is important to examine whether MFN1/2 itself is changed by the compounds in both control and disease models.

4) There is no statement about where the mitochondria are evaluated for mitochondrial transport, size, membrane potential etc.: are these in axons, dendrites, or cell bodies? What are the directions of the movement (anterograde or retrograde)? As the mechanisms underlying directional movements are different among different parts of a neuron, it is important to address those questions. For example, is the decreased % of mobile caused by less mitochondria moving in (anterograde) or more out (retrograde)? In Figure 6, the directionality of mitochondrial transport can be easily distinguished. In addition, representative images used for quantification of these parameters should be shown, otherwise it is difficult to evaluate the quality of the data. For example. "mitochondrial aspect ratio" requires a high resolution (such as under EM) and it is unclear if the immunostaining images can provide such resolution.

---

## [Author Response]

Essential revisions:1) The authors should comment on the additional role of MFN2 as a well-recognized ER-mitochondrial tether at the mitochondrial-associated ER membranes (MAM) (de Brito 2008 Nature, 456, 605., Naon 2016 PNAS., 113, 11249.). MAM is a dynamic platform in which several cellular pathways are regulated (Phospholipids synthesis, Ca2^+^, cholesterol, mito (Pera et al., 2017 EMBO J., 36, 3356-3371). Moreover, a few MAM functions are affected in CMT2A patient fibroblasts (Larrea et al., 2019) and in other sensory neuropathies described in Krols et al., 2019 HMG 28:4). Therefore, authors should demonstrate or at least discuss whether the effects of MiM111 are due to specifically to MFN1 and MFN2 mitochondrial fusion activity or might also be related to other MAM activities.

Thank you. We have added the following paragraph to the Discussion:

“As introduced above, damaging MFN2 mutations are a straightforward cause of CMT2A, but MFN2 multifunctionality complicates delineating the underlying cellular pathology (Filadi, Pendin and Pizzo, 2018; Dorn, 2020). […] Finally, MFN2 can mediate physical interactions and calcium signaling between mitochondria and endoplasmic reticulum that may also have a role in CMT2A (Larrea et al., 2019), but effects of mitofusin activation on mitochondria-reticular interactions have not been described.”

2) How do the mitofusin activators "activate mitofusins"? In Rocha et al., 2018 author found that mitofusin agonist activity was through stabilization of the fusion-permissive open conformation of endogenous normal MFN1 or MFN2 overcoming the dominant suppression of mitochondrial fusion caused by MFN2 T105M and R94Q. Does MiM111 activate mitofusins in the same way?

Thank you for permitting us to clarify these important points. Much of the desired information, including direct comparisons between Chimera C and MiM111, is reported in Dang et al., 2020. Briefly, Chimera (Cpd 2 in Dang et al., 2020) and MiM111 (Cpd 13b in Dang et al., 2020) are both small molecules that mimic the MFN activator peptides described in Franco et al., 2016 and Rocha et al., 2018. Because both compounds conform to the same pharmacophore model (see Dang et al., 2020, Figure 1), they both activate MFNs by altering protein conformation (FRET assay; Dang et al., 2020, Figure 3). We had shown in Figure 1—figure supplement 3 that neither compound affects mitochondrial morphology or inner membrane polarization status in cells lacking their mitofusin protein targets.

What is the specificity and selectivity of both activators?

Because MiM111 is being advanced as a drug for possible FDA approval, selectivity was measured against a panel of 44 different receptors and channels; MiM111 specificity is >1,000-fold for mitofusins (reported in Dang et al., 2020, Table 9). As Chimera does not have the characteristics of a drug we know it is specific for mitofusins among the dynamin family GTPases (Rocha et al., 2018), but have not performed more extensive selectivity screening.

Does it affect other mitochondrial GTPase, in addition to MFNs? The authors never evaluated how the supposed target, MFN1/2 (mRNA, protein, or activity) changes.

Unpublished experimental results rule out mitofusin activator (Chimera) effects on mitochondrial GTPase activity and MFN1 or MFN2 mRNA or protein levels.

Mitochondrial dynamics and function are a secondary readout and cannot be entirely attributed to MFNs. The current data cannot exclude the possibility that the effect is through other targets rather than MFNs. Could the authors add some controls that show how these activators increase endogenous mitofusin 'activity'? Is it due to increase in endogenous protein stability, transcription/translation or GTPase activity?

See our above response and data showing that MFN protein levels, transcription, and GTPase activity are not affected by the allosteric activators, whereas MFN conformation (and therefore fusogenicity; Franco et al., 2016) is. Perhaps the most compelling counterpoint to the argument that “mitofusin activator effects might be through other targets than mitofusins” is that the compounds have no effects in cells lacking mitofusins (Figure 1—figure supplement 3). To summarize the additional evidence that small molecule mitofusin activators exert their effects by directly interacting with MFN1 and MFN2: 1) they are highly potent (EC50 < 10nM) (Rocha et al., 2018 and Dang et al., 2018); 2) they engage the target proteins by binding to and being displaced from mitofusin HR2 domains (Rocha et al., 2018); 3) they activate MFN1 and MFN2 equally (Rocha et al., 2018, Dang et al., 2020 and Figure 1—figure supplement 3); 4) they provoke the same conformation change in MFN2 as their parent peptide (Rocha et al., 2018, Dang et al., 2020); 5) MiM111, which exists as cis- and trans isostereomers, exhibits stereoisomeric-specific activity that is not seen with non-specific interactions (Dang et al., 2020); 6) they have no non-specific effects on related dynamin family proteins or off-target effects in a Pharmaceutical Mini-Safety Panel of receptors and enzymes (Rocha et al., 2018 and Dang et al., 2020).

3) Regarding the extended used of MiM111 for all CMT2A mutations; could the authors address the fact that studies in the nervous system, fibroblast and reprogrammed motor neurons from CMT2A patients, have displayed a variety of altered mitochondrial morphology, including swelling, degeneration and altered distribution of mitochondria (Verhoeven et al., 2006, Amiott et al., 2008; Larrea et al., 2019; Saporta, et al., 2015. Exp. Neurol., 263, 190-199). However mitochondrial fragmentation was not observed. Nevertheless, all mutations caused CMT2A. Would MiM111 still be a rational therapy in all these cases? Why does the compound have no effect on control cells? As mentioned above, it is important to examine whether MFN1/2 itself is changed by the compounds in both control and disease models.

Thank you for this question. “Mitochondrial fragmentation” in CMT2A expressing cells was first reported by David Chan who expressed MFN2 mutants in murine fibroblasts (Detmer and Chan, 2007) and is also described in CMT2A patient fibroblasts by Beręsewicz, Małgorzata, Anna Boratyńska-Jasińska, Łukasz Charzewski, Maria Kawalec, Dagmara Kabzińska, Andrzej Kochański, Krystiana A. Krzyśko, and Barbara Zabłocka. (2017) The effect of a novel c. 820C> T (Arg274Trp) mutation in the mitofusin 2 gene on fibroblast metabolism and clinical manifestation in a patient. PloS one, 12(1), e0169999.

To better make this point the revised manuscript includes an additional CMT2A reprogrammed motor neuron line, MFN2 R364W that expands the number of genetically distinct CMT2A examples to 4.

Regarding iPSC-derived neurons, as suggested in the manuscript (subsection “Genetically diverse CMT2A patient neurons exhibit similar mitochondrial phenotypes”, second paragraph) our impression is that reverting fibroblasts to the iPS cell embryonic phenotype regresses the disease to its silent embryonic phenotype. We avoid this by directly reprogrammed patient fibroblasts to neurons, thereby circumventing the disease reset back to the embryo stage. Accordingly, we believe that mitofusin activation may be an attractive approach for all dominant suppressive CMT2A-causing MFN2 mutations.

4) There is no statement about where the mitochondria are evaluated for mitochondrial transport, size, membrane potential etc.: are these in axons, dendrites, or cell bodies? What are the directions of the movement (anterograde or retrograde)? As the mechanisms underlying directional movements are different among different parts of a neuron, it is important to address those questions. For example, is the decreased % of mobile caused by less mitochondria moving in (anterograde) or more out (retrograde)? In Figure 6, the directionality of mitochondrial transport can be easily distinguished. In addition, representative images used for quantification of these parameters should be shown, otherwise it is difficult to evaluate the quality of the data.

We would like to draw the reviewers’ attention to the following statements in Figure 3A legend (“mitochondrial motility in CMT2A mouse sciatic nerve axons”), Figure 5B legend (“Kymographs of mitochondrial motility in axons of live DRGs”), and the schematic depiction in Figure 6A with this description: “Yellow areas show proximal axon where mitochondrial motility was measured and distal axon where mitochondrial aspect ratio was measured”. Direction of mitochondrial transport is indicated as red vs. blue in the modified kymographs in Figures 3A and 5B with the accompanying descriptions: Figure 3A-“Bottom panel emphasize motile mitochondria with red and blue lines transiting antegrade or retrograde, respectively. (Note, mitochondrial transport in ex vivo sciatic nerves favors the antegrade [spine to foot] direction because mitochondria are recruited to the site of nerve injury at the distal amputation site [Zhou et al., 2016])”; Figure 5B- “Bottom panels emphasize motile mitochondria with red and blue lines transiting left to right or right to left, respectively”.

Unaltered greyscale representative kymographs are shown immediately above the red/blue modifications so readers can evaluate the raw data.

For example. "mitochondrial aspect ratio" requires a high resolution (such as under EM) and it is unclear if the immunostaining images can provide such resolution.

We disagree that mitochondrial aspect ratio determinations require electron microscopy. Most importantly for our work, one cannot perform electron microscopy on living cells. Except for mouse tissue histology and ultrastructure, all the data we report are from living cells, which is essential to simultaneously assessing mitochondrial morphology, polarization status and motility.

Epifluorescence microscopy has been the standard approach for analyzing mitochondrial morphology since Rizzuto and Pozzan first described its use in 1995 (R. Rizzuto, M. Brini, P. Pizzo, M. Murgia, T. Pozzan. Chimeric green fluorescent protein as a tool for visualizing subcellular organelles in living cells. Curr. Biol., 5:635-642). For a more recent example see (Koopman WJ, Visch HJ, Smeitink JA, Willems PH. Simultaneous quantitative measurement and automated analysis of mitochondrial morphology, mass, potential, and motility in living human skin fibroblasts. Cytometry A. 2006;69(1):1-12. doi:10.1002/cyto.a.20198). A PubMed search for “mitochondrial aspect ratio” yielded 258 results, of which the vast majority used fluorescence microscopy techniques. We believe our methods are rigorous and accurate; they have proven to be quite reproducible.